# Data Minimization at Inference Time

**Cuong Tran**
Department of Computer Science
University of Virginia
kxb7sd@virginia.edu

**Ferdinando Fioretto**
Department of Computer Science
University of Virginia
fioretto@virginia.edu

## Abstract

In domains with high stakes such as law, recruitment, and healthcare, learning models frequently rely on sensitive user data for inference, necessitating the complete set of features. This not only poses significant privacy risks for individuals but also demands substantial human effort from organizations to verify information accuracy. This paper asks whether it is necessary to use *all* input features for accurate predictions at inference time. The paper demonstrates that, in a personalized setting, individuals may only need to disclose a small subset of their features without compromising decision-making accuracy. The paper also provides an efficient sequential algorithm to determine the appropriate attributes for each individual to provide. Evaluations across various learning tasks show that individuals can potentially report as little as 10% of their information while maintaining the same accuracy level as a model that employs the full set of user information.

## 1 Introduction

The remarkable success of machine learning (ML) models also brought with it pressing challenges at the interface of privacy and decision-making,especially when deployed in consequential domains such as legal processes, banking, hiring, and healthcare [23]. A particularly intriguing aspect is the conventional requirement for users to disclose their entire set of features during inference, thereby creating a gateway for potential data breaches, as exemplified in recent instances where millions of individuals' data were compromised []. Concurrently, this practice also places a burden on companies and organizations to ensure the accuracy and legal compliance of the disclosed information, as often observed in financial operations, mandated by legislations like the Corporate Transparency Act[].

Significantly, this conventional approach of disclosing the enture feature set during inference might also also violate the data minimization principle, a cornerstone of several global privacy regulations including the Eurpean General Data Protection Regulation [2], the California Privacy Rights Act [1], and the Brazilian General Data Protection Law [3], among others. Through this lens, the discourse on enhancing privacy and reducing the verification onus in ML systems, particularly in crucial decision-making domains, gains a nuanced dimension, warranting a thorough examination.

This paper challenges this setting and asks whether it is necessary to require *all* input features for a model to produce accurate or nearly accurate predictions during inference. We refer to this question as the *data minimization for inference* problem. This unique question bears profound implications for privacy in model personalization, which often necessitates the disclosure of substantial user data. We show that, under a personalized setting, each individual may only need to release a small subset of their features to achieve the *same* prediction errors as those obtained when all features are disclosed. The overall framework is depicted in Figure 1. Following this result, we also provide an efficient sequential algorithm to identify the minimal set of attributes that each individual should reveal. Evaluations across various learning tasks indicate that individuals may be able to report as little as 10% of their information while maintaining the same accuracy level as a model using the complete set of user information.

37th Conference on Neural Information Processing Systems (NeurIPS 2023).

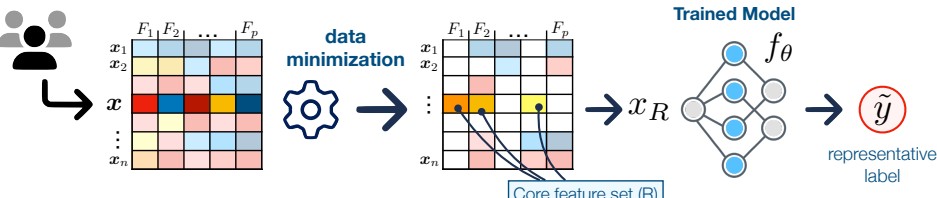

Figure 1: Illustrative example of the proposed framework. The figure higlights the *online* and *personalized* nature of the minimization process. For each user, a key subset of features is identified to be used (core feature set). The minimized data sample is then processed by the pre-trained model, which generates a *representative* label.

In summary, the paper makes the following contributions: **(1)** it initiates a study to analyze the optimal subset of data features that each individual should disclose at inference time in order to achieve the same accuracy as if all features were disclosed; **(2)** it links this analysis to a new concept of *data minimization for inference* in relation to privacy, **(3)** it proposes theoretically motivated and efficient algorithms for determining the minimal set of attributes each individual should provide to minimize their data; and **(4)** it conducts a comprehensive evaluation illustrating the effectiveness of the proposed methods in preserving privacy at no or small costs in accuracy.

**Related work.** To the best of our knowledge, this is the first work studying the connection between data minimization and accuracy at inference time. This work was motivated by the core principle in data minimization which states that *"all collected data shall be adequate, relevant and limited to what is necessary in relation to the purposes for which they are processed"*. This principle is adopted as a privacy cornerstone in different data protection regulations, including the General Data Protection Regulation [2] in Europe, the California Privacy Rigths Act in the US [1], the General Personal Data Protection Law in Brasil [3], the Protection of Personal Information Act, in South Africa [5], and the Personal Information Protection Act, in South Korea [4].

While we are not aware of studies on data minimization for inference problems, we draw connections with differential privacy, feature selection, and active learning. *Differential Privacy (DP)* [14] is a strong privacy notion that determines and bounds the risk of disclosing sensitive information of individuals participating in a computation. In the context of machine learning, DP ensures that algorithms can learn the relations between data and predictions while preventing them from memorizing sensitive information about any specific individual in the training data. In such a context, DP is primarily adopted to protect training data [6, 12, 32] and thus the setting contrasts with that studied in this work, which focuses on identifying the superfluous features revealed by users at *test time* to attain high accuracy. Furthermore, achieving tight constraints in differential privacy often comes at the cost of sacrificing accuracy, while the proposed privacy framework can reduce privacy loss without sacrificing accuracy under the assumption of linear classifiers.

Feature selection [10] is the process of identifying a relevant subset of features from a larger set for use in model construction, with the goal of improving performance by reducing the complexity and dimensionality of the data. The problem studied in this work can be considered a specialized form of feature selection with the added consideration of personalized levels, where each individual may use a different subset of features. This contrasts standard feature selection [20], which selects the same subset of features for each data sample. Additionally, and unlike traditional feature selection, which is performed during training and independent of the deployed classifier [10], the proposed framework performs feature selection at deployment time and is inherently dependent on the deployed classifier.

The framework proposed in this paper shares some similarities with *active learning* [15, 26, 33], whose goal is to iteratively select samples for experts to label in order to construct an accurate classifier with the least number of labeled samples. Similarly, the proposed framework iteratively asks individuals to reveal one attribute given their released features so far, with the goal of minimizing the uncertainty in model predictions. Finally, our study share some connections with active feature acquisition [28, 13, 11, 21] which can be categorized as one branch in active learning. However, these priors focus mostly on minimizing the total number of users features at training step. This is in sharp contrast with ours work which concentrate on inferece time.

## 2 Settings and objectives

We consider a dataset $D$ consisting of samples $(x, y)$ drawn from an unknown distribution $\Pi$. Here, $x \in \mathcal{X}$ is a feature vector and $y \in \mathcal{Y} = [L]$ is a label with $L$ classes. The features in $x$ are categorized into *public* $x_P$ and *sensitive* $x_S$ features, with their respective indexes in vector $x$ denoted as $P$ and $S$, respectively. We consider classifiers $f_\theta : \mathcal{X} \to \mathcal{Y}$, which are trained on a public dataset from the same data distribution $\Pi$ above. The classifier produces a score $\tilde{f}_\theta(x) \in \mathbb{R}^L$ over the classes and a final output class, $f_\theta(x) \in [L]$, given input $x$. The model's outputs $f_\theta(x)$ and $\tilde{f}_\theta(x)$ are also often referred to as hard and soft predictions, respectively.

Without loss of generality, we assume that all features in $\mathcal{X}$ lie within the range $[-1, 1]$. In this setting, we are given a trained model $f_\theta$ and, at inference time, we have access to the public features $x_P$. These features might be revealed through user queries or collected by the provider during previous interactions. Our focus is on the setting where $|S| \ll |D|$, and for simplicity, the following considers binary classifiers, where $\mathcal{Y} = \{0, 1\}$ and $\tilde{f}_\theta \in \mathbb{R}$. Multi-class settings are addressed in Appendix C.

In this paper, the term **data leakage** of a model, refers to the percentage of sensitive features that are revealed unnecessarily, meaning that their exclusion would not significantly impact the model's output. *Our goal is to design algorithms that accurately predict the output of the model using the smallest possible number of sensitive features*, *thus minimizing the data leakage at inference time*. This objective reflects our desire for privacy.

To clarify key points discussed in the paper, let us consider a loan approval task where individual features are represented by the set $\{Job, Loc(action), Inc(ome)\}$. In this example, the *Job* feature $\in x_P$ is public, whereas *Loc* and *Inc* $\in x_S$ are sensitive. We also consider a trained linear model $f_\theta = 1.0\, Job - 0.5\, Loc + 0.5\, Inc \geq 0$ and look at a scenario where user (A) has a public feature $Job = 1.0$, and user (B) has a public feature $Job = -0.9$. Both users' sensitive feature values are unknown. However, for user A, the outcome can be conclusively determined without revealing any additional information since all features are bounded within $[-1, 1]$. In contrast, for user B, the outcome cannot be determined solely based on the public feature, but revealing the sensitive feature $Loc = 1.0$ is enough to confirm the classifier outcome.

This example highlights two important observations that motivate our study: **(1)** *not all sensitive attributes may be required for decision-making during inference*, and **(2)** *different individuals may need to disclose different amounts and types of sensitive information for decision-making*.

## 3 Core feature sets

With these considerations in mind, this section introduces the notion of *core feature set*, the *first contribution of the paper*, which will be used to quantify data minimization. The paper presents the key findings and defers all proofs in Appendix A.

Throughout the paper, the symbols $R$ and $U$ are used to represent the sets of indices for revealed and unrevealed features of the sensitive attribute $S$, respectively. Given a vector $x$ and an index set $I$, we use $x_I$ to denote the vector containing entries indexed by $I$ and $X_I$ to represent the corresponding random variable. Finally, we write $f_\theta(X_U, X_R = x_R)$ as a shorthand for $f_\theta(X_U, X_R = x_R, X_P = x_P)$ to denote the prediction made by the model when the features in $U$ are unrevealed.

*Our objective is to develop algorithms that can identify the smallest subset of sensitive features to disclose*, ensuring that the model's output is accurate (with high probability) irrespective of the values of the undisclosed features. We refer to this subset as the *core feature set*.

**Definition 1** (Core feature set). *Consider a subset $R$ of sensitive features $S$, and let $U = S \setminus R$ be the unrevealed features. The set $R$ is a core feature set if, for some $\tilde{y} \in \mathcal{Y}$,*

$$\Pr\left(f_\theta(X_U, X_R = x_R) = \tilde{y}\right) \geq 1 - \delta, \tag{1}$$

*where $\delta \in [0, 1]$ is a failure probability.*

When $\delta = 0$ the core feature set is called **pure**. Additionally, the label $\tilde{y}$ satisfying Equation (1) is called the **representative label** for the core feature set $R$. The concept of the representative label $\tilde{y}$ is crucial for the algorithms that will be discussed later. These algorithms use limited information to

make predictions and when predictions are made using a set of unrevealed features, the representative label $\tilde{y}$ will be used in place of the model's prediction.

In identifying core feature sets to minimize data leakage, it's crucial to consider model uncertainty, which refers to the unknown values of unrevealed features. The following result links core feature sets with model entropy, which measures uncertainty and is used by this work to minimize data leakage.

**Proposition 1.** *Let $R \subseteq S$ be a core feature set with failure probability $\delta < 0.5$. Then, there exists a monotonic decreasing function $\epsilon : \mathbb{R}_+ \to \mathbb{R}_+$ with $\epsilon(1) = 0$ such that:*

$$H\big[f_\theta(X_U, X_R = x_R)\big] \leq \epsilon(1 - \delta),$$

*where $H[Z] = -\sum_{z \in [L]} \Pr(Z = z) \log \Pr(Z = z)$ is the entropy of the random variable $Z$.*

This property *highlights the relationship between core feature sets and entropy associated with a model using incomplete information.* As the $\delta$ value decreases, the model's predictions become more certain. When $\delta$ equals zero (or when $R$ represents a pure core feature set), the model's predictions can be fully understood without observing $x_U$, resulting in entropy of $0$.

It is worth noticing that enhancing prediction accuracy necessitates revealing additional information, as illustrated by the previous result and the renowned information theoretical proposition below:

**Proposition 2.** *Given two subsets $R$ and $R'$ of sensitive features $S$, with $R \subseteq R'$,*

$$H\big(f_\theta(X_U, X_R = x_R)\big) \geq H\big(f_\theta(X_{U'}, X_{R'} = x_{R'})\big),$$

*where $U = S \setminus R$ and $U' = S \setminus R'$.*

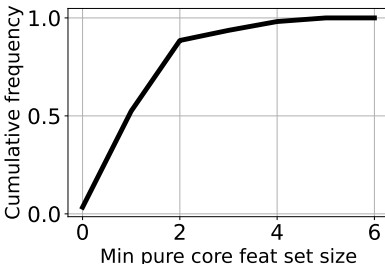

Thus, the parameter $\delta$ plays a crucial role in balancing the trade-off between *privacy loss* and *model prediction's uncertainty*. It determines the amount of sensitive information that must be disclosed to have enough confidence on model predictions. As $\delta$ increases, fewer sensitive features need to be revealed, resulting in reduced data leakage but also less certainty on model predictions, and vice versa. Note that disclosing more sensitive attributes *does not necessarily improve the accuracy of predictions for a specific individual*, as highlighted in previous research [30]. Although it's challenging to rigorously analyze the relationship between privacy loss and predictive accuracy across various contexts, our empirical studies across multiple datasets and learning algorithms do suggest that exposing more sensitive features generally enhances prediction accuracy. Thus, the parameter $\delta$ appears to govern the trade-off between sacrificing privacy and gaining predictive performance.

Figure 2: Frequency associated with the size of the **minimum** pure core feature set.

As highlighted in the previous example, the core feature set is not uniform for all users. This is further exemplified in Figure 2, using the Credit dataset [9] with a logistic regression classifier. The figure reports the cumulative count of users against the minimum number of features they need to disclose to ensure confident predictions. It demonstrates that many individuals need to disclose *no* additional information to attain accurate predictions (corresponding to a pure feature of set size $0$), and most individuals can achieve accurate predictions by disclosing only $\leq 2$ sensitive features. These insights, together with the previous observations linking core feature sets to entropy, motivate the proposed online algorithm, the second contribution of the paper.

## 4 MinDRel: An algorithm to minimize data release at inference time

The goal of the proposed algorithm, called *Minimize Data Reveal* (MinDRel), is to uphold privacy during inference by revealing sensitive features one at a time based on their *released* feature values. This section provides a high-level description of the algorithm and outlines its challenges. Next, Section 5, applies MinDRel to linear classifiers and discusses its performance on several datasets and benchmarks. Further, Section 6, extends MinDRel to non-linear classifiers and considers an evaluation over a range of standard datasets.

**Overview of MinDRel.** MinDRel operates fundamentally on two critical actions: *1. determining the next feature to reveal* for each user and *2. verifying whether the disclosed features make up a core feature set* for that user. These two operations will be discussed in sections 4.1 and 4.2, respectively.

The algorithm determines which feature to disclose for a specific user by inspecting the posterior probabilities $\Pr(X_j|X_R = x_R, X_P = x_P)$ for each unrevealed feature $j \in U$, taking into account the disclosed sensitive features $x_R$ and public features $x_P$. Given the current set of disclosed features $x_R$ and unrevealed features $x_U$, MinDRel chooses the subsequent feature $j \in U$ as follows:

$$j = \underset{j \in U}{\operatorname{argmax}} \, F(x_R, x_j; \theta) \doteq \underset{j \in U}{\operatorname{argmax}} -H\big[f_\theta(X_j = x_j, X_{U \setminus \{j\}}, X_R = x_R)\big], \qquad (2)$$

where $F$ is a *scoring function* that evaluates the amount of information that can be acquired about the model's predictions when feature $X_j$ is disclosed. As suggested in previous sections, it's desirable to reveal the feature that provides the most insight into the model prediction upon disclosure. MinDRel uses *Shannon entropy* for this purpose as it offers a natural method for quantifying information. Once feature $X_j$ is disclosed with a value of $x_j$, the algorithm updates the posterior probabilities for all remaining unrevealed features. The process concludes either when all sensitive features have been disclosed or when a core feature set has been identified. It should also be noted that, within this framework, there is no need to perform data imputation when some features are missing. Unrevealed features are treated as random variables and are integrated during the prediction process.

Both the computation of the scoring function $F$ and the verification of whether a set of disclosed features constitute a core feature set present two significant challenges for the algorithm. The rest of the section delves into these difficulties.

## 4.1 Computing the scoring function $F$

Designing a scoring function $F$ that measures how confident a model's prediction is when a user discloses an additional feature $X_j$ brings up two key challenges. **First**, *the value of $X_j$ is unknown until the decision to reveal it is made*, which complicates the computation of the entropy function. **Second**, even if the value of $X_j$ were known, *determining the entropy of model predictions in an efficient manner* is another difficulty. We next discuss how to overcome these challenges.

**Dealing with unknown values.** To address the first challenge, we exploit the information encoded in the disclosed features to infer $X_j$'s value and compute the posterior probability $\Pr(X_j|X_R = x_R)$ of the unrevealed feature $X_j$ given the values of the revealed ones. The scoring function, abbreviated as $F(X_j)$, can thus be modeled as the expected negative entropy given the randomness of $X_j$,

$$F(X_j) = \mathbb{E}_{X_j} - \big[H[f_\theta(X_j, X_{U \setminus \{j\}}, X_R = x_R)]$$
$$= -\int \underbrace{H\big[f_\theta(X_j = z, X_{U \setminus \{j\}}, X_R = x_R)\big]}_{A} \times \underbrace{\Pr(X_j = z|X_R = x_R)}_{B} \, dz, \qquad (3)$$

where $z \in \mathcal{X}_j$ is a value in the support of $X_j$.

**Efficient entropy computation.** The second difficulty relates to how to estimate this scoring function efficiently. Indeed this is challenged by two key components. The first (A) is the entropy of the model's prediction given a specific unrevealed feature value, $X_j = z$. This prediction is a function of the random variable $X_{U \setminus \{j\}}$, and, due to Proposition 1, its estimation is linked to the conditional densities $\Pr(X_{U \setminus \{j\}}|X_R = x_R, X_j = z)$. The second (B) is the conditional probability $Pr(X_j = z|X_R = x_R)$. Efficient computation of these conditional densities is discussed next.

First, we discuss a result relying on the joint Gaussian assumption of the input features. This result will be useful in providing a computationally efficient method to estimate such conditional density functions. In the following, $\Sigma_{IJ}$ represents a sub-matrix of size $|I| \times |J|$ of a matrix $\Sigma$ formed by selecting rows indexed by $I$ and columns indexed by $J$.

**Proposition 3.** *The conditional distribution of any subset of unrevealed features $U' \subseteq U$, given the the values of released features $X_R = x_R$ is given by:*

$$\Pr(X_{U'}|X_R = x_R) = \mathcal{N}\bigg(\mu_{U'} + \Sigma_{U'R}\Sigma_{RR}^{-1}(x_R - \mu_R), \Sigma_{U'U'} - \Sigma_{U'R}\Sigma_{RR}^{-1}\Sigma_{RU'}\bigg),$$

*where $\Sigma$ is the covariance matrix.*

Note that Equation (3) considers $U' = \{j\}$, and thus, component (B) can be computed efficiently exploiting the result above. To complete Equation (3), we need to estimate the

entropy $H[f_\theta(X_j = z, X_{U \setminus j}, X_R = x_R)]$ (component A) for a specific instance $z$ drawn from $\Pr(X_j | X_R = x_R)$. This poses a challenge due to the non-linearity of the hard model predictions $f_\theta$ adopted. To tackle this computational challenge, we first estimate component A using soft labels $\tilde{f}_\theta$ and then apply a thresholding operator. More specifically, we first estimate $\Pr(\tilde{f}_\theta(X_j = z, X_{U \setminus \{j\}}, X_R = x_R))$ and, based on this distribution, we subsequentially estimate $f_\theta$ as $\mathbf{1}\{\tilde{f}_\theta \geq 0\}$, where $\mathbf{1}$ is the indicator function. In the following sections, we will show how to assess this estimate for linear and non-linear classifiers. Finally, by approximating the distribution over soft model predictions through Monte Carlo sampling, the score function in $F(X_j)$ can be computed as

$$F(X_j) \approx -1/|\mathbf{Z}| \sum_{z' \in \mathbf{Z}} H\left[f_\theta(X_j = z', X_{U \setminus \{j\}}, X_R = x_R)\right], \tag{4}$$

where $\mathbf{Z}$ is a set of random samples drawn from $\Pr(X_j | X_R = x_R)$ and estimated through Proposition 3, which thus can be computed efficiently.

When the Gaussian assumption does not hold, one can recur to (slower) Bayesian approaches to estimate the uncertainty of unrevealed features $X_U$ given the set of revealed features $X_R = x_R$. A common approach involves treating $X_U$ as the target variable and employing a neural network to establish the mapping $X_U = g_w(X_R)$. Utilizing Bayesian techniques [17, 18], the posterior of the network's parameter $p(w|D) = p(w)p(D|w)$ can be computed initially. Based on the posterior distribution of the model's parameters $w$, the posterior of unrevealed features can be calculated as $\Pr(X_U = x_U | X_R = x_R) = \Pr_{w \sim p(w|D)}(g_w(X_R) = x_U)$. However, implementing such a Bayesian network not only significantly increases training time but also inference time. Since it is necessary to compute $\Pr(X_U | X_R = x_R)$ for all possible choices of $U \in S$, the number of Bayesian neural network regressors scales exponentially with $|S|$.

Importantly, in our evaluation, the data mini-mization method that operates under the Gaussian assumption maintains similar decision-making and produces comparable outcomes to the Bayesian approach, even in cases where the Gaussian assumption is not applicable in practical settings. Figure 3 illustrates this comparison, showcasing the performance of the proposed mechanism on a real dataset (Credit dataset with

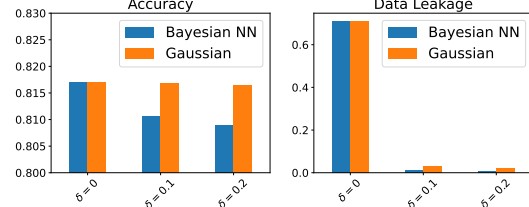

Figure 3: Comparison between Bayesian NN vs Gaussian in term of accuracy and data Lekage.

$|S| = 5$) concerning accuracy (higher is better) and data leakage (lower is better) across various failure probability $\delta$ values. Notice how similar is the performance of the mechanisms that either leverage the Gaussian assumption or operate without it (Bayesian NN). Importantly, the assumption of a Gaussian distribution is not overly restrictive or uncommon. In fact, it is a cornerstone in many areas of machine learning, including Gaussian Processes [31], Bayesian optimization [27], and Gaussian Graphical models [24].

### 4.2 Testing a core feature set

The proposed iterative algorithm terminates once it determines whether a subset $R$ of the sensitive feature set $S$ constitutes a core feature set. This validation process falls into two scenarios:

1. When $\delta = 0$: To confirm that $R$ is a pure core feature set, it is sufficient to verify that $f_\theta(X_U, X_R = x_R)$ remains constant for all possible realizations of $X_U$. As we will show in Section 5, linear classifiers can perform this check in linear time without making any specific assumptions about the input distribution.

2. When $\delta > 0$: In this case, the property above is no longer valid. As per Definition 1, to confirm a core feature set, it is essential to estimate the distribution of $\Pr(\tilde{f}_\theta(X_U, X_R = x_R))$. In Section 5, we demonstrate how to analytically estimate this distribution for linear classifiers. Furthermore, in Section 6, we illustrate how to locally approximate this distribution for nonlinear classifiers and derive a simple yet effective estimator that can be readily implemented in practice.

## 5 MinDRel for linear classifiers

This section will devote to estimating the distribution $\Pr(\tilde{f}_\theta(X_j = z, X_{U \setminus \{j\}}, X_R = x_R))$, simply expressed as $\Pr(\tilde{f}_\theta(X_U, X_R = x_R))$ and provides an instantiation of MinDRel for linear classifiers.

In particular, it shows that both the estimation of the conditional distributions required to compute the scoring function $F(X_j)$ and the termination condition to test whether a set of revealed features is a core feature set, can be computed efficiently. This is an important property for the developed algorithms, which are designed to be online and interactive.

## 5.1 Efficiently Estimating $\Pr(\tilde{f}_\theta(X_U, X_R = x_R))$

For a linear classifier $\tilde{f}_\theta = \theta^\top x$, notice that when the input features are jointly Gaussian, the model predictions $\tilde{f}_\theta(x)$ are also Gaussian, as highlighted by the following result.

**Proposition 4.** *The model soft prediction, $\tilde{f}_\theta(X_U, X_R = x_R) = \theta_U X_U + \theta_R x_R$ is a random variable following a Gaussian distribution $\mathcal{N}(m_f, \sigma_f^2)$, with*

$$m_f = \theta_R x_R + \theta_U^\top (\mu_U + \Sigma_{UR}\Sigma_{RR}^{-1}(x_R - \mu_R)) \tag{5}$$

$$\sigma_f^2 = \theta_U^\top (\Sigma_{UU} - \Sigma_{UR}\Sigma_{RR}^{-1}\Sigma_{RU})\theta_U, \tag{6}$$

*where $\theta_U$ is the sub-vector of parameters $\theta$ corresponding to the unrevealed features $U$.*

The result above is used to assist in calculating the conditional distribution of the model hard predictions $f_\theta(x)$, following thresholding. This is a random variable that adheres to a Bernoulli distribution, as shown next, and will be used to compute the entropy of the model predictions, as well as to determine if a subset of features constitutes a core set.

**Proposition 5.** *Let the soft model predictions $\tilde{f}_\theta(X_U, X_R = x_R)$ be a random variable following a Gaussian distribution $\mathcal{N}(m_f, \sigma_f^2)$. Then, the model prediction following thresholding $f_\theta(X_U, X_R = x_R)$ is a random variable following a Bernoulli distribution $\mathcal{B}(p)$ with $p = \Phi(\frac{m_f}{\sigma_f})$, where $\Phi(\cdot)$ is the CDF of the standard Normal distribution, and $m_f$ and $\sigma_f$, are given in Eqs (5) and (6), respectively.*

## 5.2 Testing pure core feature sets

In this subsection, we outline the methods for determining if a subset $U$ is a pure core feature set, and, if so, identifying its representative label. As per Definition 1, $U$ is a pure core feature set if $f_\theta(X_U, X_R = x_R) = \tilde{y}$ for all $X_U$. This implies that $\tilde{f}_\theta(X_U, X_R = x_R) = \theta_U^\top X_U + \theta_R^\top x_R$ must have the same sign for all $X_U$ in the range of $[-1, 1]^{|U|}$. Given the box constraint $X_U \in [-1, 1]^{|U|}$, the linearity of the model considered allows us to directly compute the maximum and minimum values of $\tilde{f}_\theta(X_U, X_R = x_R)$, rather than enumerating all possible values. Specifically, we have:

$$\max_{X_U} \theta_U^\top X_U + \theta_R^\top x_R = \|\theta_U\|_1 + \theta_R^\top x_R$$

$$\min_{X_U} \theta_U^\top X_U + \theta_R^\top x_R = -\|\theta_U\|_1 + \theta_R^\top x_R.$$

Thus, if both these maximum and minimum values are negative (non-negative), then $U$ is considered a pure core feature set with representative label $\tilde{y} = 0$ ($\tilde{y} = 1$). If not, $U$ is not a pure core feature set.

Importantly, determining whether a subset $R$ of sensitive features $S$ constitutes a pure core feature set can be accomplished in linear time with respect to the number of features.

**Proposition 6.** *Assume $f_\theta$ is a linear classifier. Then, determining if a subset $U$ of sensitive features $S$ is a* pure *core feature set can be performed in $O(|P| + |S|)$ time.*

## 5.3 MinDRel-linear Algorithm and Evaluation

A pseudo-code of MinDRel specialized for linear classifiers is reported in Algorithm 1. At inference time, the algorithm takes as input a sample $x$ (which only exposes the set of public features $x_P$) and uses the training data $D$ to estimate the mean and covariance matrix needed to compute the conditio-

nal distribution of the model predictions given the unrevealed features (line 1), as discussed above. After initializing empty the set of revealed features (line 2), it iteratively releases a feature at a time until a core feature set (and its associated representative label) are determined, as discussed in detail in Section 5.2. The released feature $X_{j^*}$ is the one, among the unrevealed features $U$, that maximizes the scoring function $F$ (line 12). Computing such a scoring function entails estimating the conditional

distribution $\Pr(X_j | X_R = x_R)$ (line 8), constructing a sample set $\mathbf{Z}$ from such distribution (line 9), and approximating the distribution over soft model predictions through Monte Carlo sampling to compute (line 10). Finally, after each iteration, the algorithm updates the set of the revealed and unrevealed features (line 13).

Notice that MinDRel relies on estimating the mean vector and covariance matrix from the training data, which is considered public, for the scope of this paper. If the training data is private, various techniques exist to release DP mean, and variance [22, 7] and can be readily adopted. Nonetheless, the protection of training data is beyond the focus of this work.

---

**Algorithm 1:** MinDRel for linear classifiers

**input** : A test sample $x$; training data $D$
**output** : A core feature set $R$ and its representative label $\tilde{y}$

1  $(\mu, \Sigma) \leftarrow \left( \frac{1}{|D|} \sum_{(x,y) \in D} x, \frac{1}{|D|} \sum_{(x,y) \in D} (x - \mu)(x - \mu)^\top \right)$
2  Initialize $R = \emptyset$
3  **while** *True* **do**
4      **if** $R$ *is a core feature set with repr. label* $\tilde{y}$ **then**
5          **return** $(R, \tilde{y})$
6      **else**
7          **foreach** $j \in U$ **do**
8              Compute $\Pr(X_j | X_R = x_R)$ (using Prop. 3)
9              $\mathbf{Z} \leftarrow$ sample($\Pr(X_j | X_R = x_R)$) T times
10             Compute $\Pr\left( f_\theta(X_j = z, X_{U \setminus \{j\}} X_R = x_R) \right)$ (using Prop. 4 and 5)
11             Compute $F(X_j)$ (using Eq. (4))
12     $j^* \leftarrow \operatorname{argmax}_j F(X_j)$
13     $(R, U) \leftarrow R \cup \{j^*\},\ U \setminus \{j^*\}$

---

Figure 4 reports the cumulative count of users (y-axis) against the minimum number of features they need to disclose (x-axis) to ensure confident predictions, for various failure probabilities $\delta$. The model adopted is a Logistic Regression classifier trained on the Bank dataset [9]. The data minimization achieved by MinDRel is clearly apparent. For each test sample, MinDRel identifies core feature sets that are significantly smaller than the total sensitive feature set size $|S| = 7$. Interestingly, when $\delta > 0$, it discovers core feature sets smaller than 2 for the majority of users. *This implies that most users would only need to reveal a small portion of their sensitive data to achieve accurate model predictions with either absolute certainty or high confidence.*

## 6 MinDRel for non-linear classifiers

Next, the paper focuses on computing the estimate $\Pr(\tilde{f}_\theta(X_U, X_R = x_R))$ and determining core feature sets when $f_\theta$ is a nonlinear classifier. Then, the section presents results that illustrate the practical benefits of MinDRel for data minimization at inference time on neural networks.

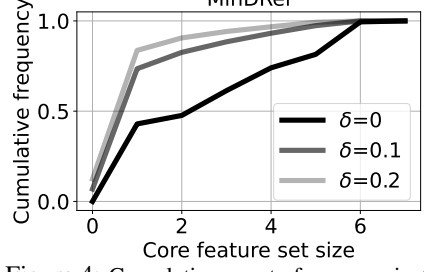

Figure 4: Cumulative count of users against their core feature set size for various $\delta$.

### 6.1 Efficiently estimating $Pr(\tilde{f}_\theta(X_U, X_R = x_R))$

First notice that even if the input features $x$ are jointly Gaussian, the outputs $f_\theta(x)$ will no longer adhere to a Gaussian distribution after a non-linear transformation. This complicates estimating the distribution $\Pr(\tilde{f}_\theta(X_U, X_R = x_R))$. To tackle this challenge, the paper takes a local approximation of the model predictions $\tilde{f}_\theta$ using a Gaussian distribution, as demonstrated in the following result.

**Theorem 1.** *The distribution of the random variable* $\tilde{f}_\theta = \tilde{f}_\theta(X_U, X_R = x_R)$ *where* $X_U \sim \mathcal{N}\left(\mu_U^{pos}, \Sigma_U^{pos}\right)$ *can be approximated by a Normal distribution as*

$$\tilde{f}_\theta \sim \mathcal{N}\left( \tilde{f}_\theta(X_U = \mu_U^{pos}, X_R = x_R),\ g_U^\top \Sigma_U^{pos} g_U \right), \tag{7}$$

*where* $g_U = \nabla_{X_U} \tilde{f}_\theta(X_U = \mu_U^{pos}, X_R = x_R)$ *is the gradient of model prediction at* $X_U = \mu_U^{pos}$.

Therein, the mean vector $\mu_U^{pos}$ and covariance matrix $\Sigma_U^{pos}$ of $Pr(X_U | X_R = x_R)$ are derived from Proposition 3. This result leverages a first-order Taylor approximation of model $f_\theta$ around its mean.

## 6.2 Testing pure core feature sets

Unlike linear classifiers, the case for nonlinear models lacks an exact and efficient method to determine whether a set is a core feature set. This is primarily due to the non-convex nature of the adopted models, which poses challenges in finding a global optimum. This section thus proposes an approximate testing routine and demonstrates its practical ability to significantly minimize data leakage during testing while maintaining high accuracy.

To determine if a subset $U$ of the sensitive features $S$ is a pure core feature set, we consider a set $Q$ of input points $(X_U, x_R)$. The entries corresponding to the revealed features are set to the value $x_R$, while the entries corresponding to the unrevealed features are sampled from the distribution $\Pr(X_U | X_R = x_R)$. The test verifies if the model predictions $f_\theta(x)$ remain constant for all $x$ in $Q$. In the next section, we will show that even considering arbitrary classifiers (e.g., we use standard neural networks), MinDRel can reduce data leakage dramatically when compared to standard approaches. The MinDRel algorithm for nonlinear classifiers differs from Algorithm 1 primarily in the method used to compute the estimates for the distribution $\Pr(f_\theta(X_j = z, X_{U \setminus j}, X_R = x_R))$ of the soft model predictions (line 11). In this case, this estimate is computed by leveraging the results from Theorem 1 and Proposition 5. Moreover, the termination test of the algorithm is based on the discussion in the previous section. Appendix B reports a description of the algorithm's pseudocode.

# 7 Experiments

**Datasets and settings.** This section evaluates MinDRel's effectiveness in limiting data exposure during inference. The experiments are conducted on six standard UCI datasets [9]. To further emphasize the benefits of MinDRel, we compare it to two baselines: the *optimal* model, which employs a brute force method to find the smallest core feature set and its representative label and assumes all sensitive features are known, and the *all-features* model, which simply adopts the original classifiers using all the data features for each test sample. The performances are displayed for a varying number of sensitive attributes $|S| \in [2, 5]$, while we delegate a study for larger $|S|$ to the Appendix (which excludes the time-consuming baseline *optimal*, as intractable for large $|S|$). For each choice of $|S|$, we randomly select $|S|$ features from the entire feature set and designate them as sensitive attributes. The remaining attributes are considered public. The average accuracy and data leakage are then reported based on 100 random sensitive attribute selections. Due to page limit, we present a selection of the results and discuss their trends on Bank dataset. A comprehensive overview, additional analysis, and experiments are available in the Appendix D.

## 7.1 Linear classifiers

Figure 5 depicts performance results in accuracy (top subplots/higher is better) and relative data leakage (bottom subplots/lower is better) with varying number of revealable sensitive features $|S|$. The comparison includes three MinDRel versions: *F-Score*, which utilizes the scoring function elaborated in Section 4.1 to select the next feature to disclose (left); *Importance*, which employs a feature importance criterion leveraging the model $f_\theta$ parameters to rank features (middle), detailed in the Appendix; and *Random*, which arbitrarily selects the next feature to reveal (right). All three versions use the same test procedure to validate whether a set qualifies as a core feature set.

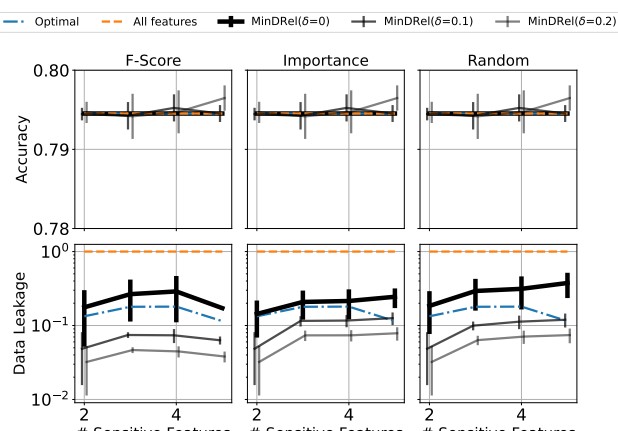

Figure 5: Comparison of F-Score, Importance and Random method for different $\delta$ with two baseline methods: Optimal and All features in term of Accuracy and Data Leakage. The underlying classifier is a logistic regression classifier.

Firstly, notice that MinDRel achieves equal or better accuracy than the optimal mechanism and baseline that utilizes all features during testing. *This suggests that data minimization can be accomplished under linear models without compromising accuracy!* Next, observe that an increase in $\delta$

aids in safeguarding data minimization, as evidenced by the drop in relative data leakage (note the logarithmic scale of the y-axis). This is attributable to the influence of $\delta$ on the test for identifying a core feature set, thereby reducing its size.

Finally, the proposed scoring function outperforms other versions in terms of data leakage minimization, allowing users to disclose substantially fewer sensitive features. The Appendix also includes experiments with larger quantities of sensitive features, presenting analogous trends, where, however, a comparison against an optimal baseline was not possible in due to its exponential time complexity.

## 7.2 Non-linear classifiers

To assess MinDRel's performance in reducing data leakage when employing standard nonlinear classifiers, we use a neural network with two hidden layers and ReLU activation functions and train the models using stochastic gradient descent (see Appendix D for additional details). The evaluation criteria, baselines, and benchmarks adhere to the same parameters set in Section 5.3.

Figure 6 showcases the results in terms of accuracy (top subplots) and data leakage (bottom subplots). Unlike linear classifiers, nonlinear models using MinDRel with a failure probability of $\delta = 0$ cannot guarantee the

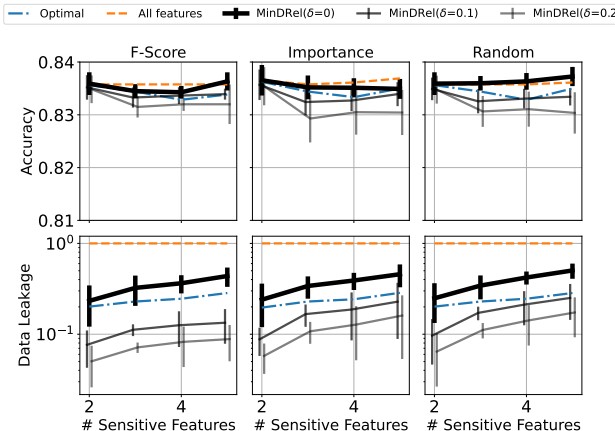

Figure 6: Comparison of F-Score, Importance and Random method for different $\delta$ with two baseline methods: Optimal and All features in term of Accuracy and Data Leakage. The underlying classifier is a neural network.

same level of accuracy as the "*all features*" baseline model. However, this accuracy difference is minimal. Notably, a failure probability of $\delta = 0$ enables users to disclose less than half, and up to 90% fewer sensitive features across different datasets, while achieving accuracies similar to those of conventional classifiers. Next, similarly to as observed in the previous section, MinDRel with the proposed F-score selector significantly outperforms the other variants in terms of data leakage minimization. Furthermore, when considering higher failure probabilities, data leakage decreases significantly. For instance, with $\delta \leq 0.1$, users need to disclose only 5% of their sensitive features while maintaining accuracies comparable to the baseline models (the largest accuracy difference reported was 0.005%). These results are significant: *They show that the introduced notion of privacy leakage and the proposed algorithm can become a valuable tool to protect individuals' data privacy at test time, without significantly compromising accuracy.*.

## 8 Conclusion and Future Work

This paper introduced the concept of data minimization at test time whose goal is to minimize the number of features that individuals need to disclose during model inference while maintaining accurate predictions from the model. The motivations of this notion are grounded in the privacy risks imposed by the adoption of learning models in consequential domains, and align with the data minimization principle. The paper then discusses an iterative and personalized algorithm that selects the features each individual should release with the goal of minimizing data leakage while retaining exact (in the case of linear classifiers) or high (for non-linear classifiers) accuracy. Experiments over a range of benchmarks and datasets indicate that individuals may be able to release as little as 10% of their information without compromising the accuracy of the model, providing a strong argument for the effectiveness of this approach in protecting privacy while preserving the accuracy of the model.

While this study is the first attempt at defining data minimization during inference, it also opens up avenues for further research. First, establishing bounds on the data leakage provided by our proposed method compared to an optimal procedure presents an interesting and open challenge. Second, exploring the relationship between data minimization principles and their consequent disparate impacts presents another open direction. Lastly, developing effective algorithms to provably construct core feature sets for non-linear classifiers is another important area of investigation.

## Acknowledgements

This research was partially supported by NSF grant 2133169, a Google Research Scholar Award and an Amazon Research Award.

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

## A Missing proofs

**Proposition 1.** *Given a core feature set $R \subseteq S$ with failure probability $\delta < 0.5$, then there exists a function $\epsilon : \mathbb{R} \to \mathbb{R}$ that is monotonic decreasing function with $\epsilon(1) = 0$ such that:*

$$H\big[f_\theta(X_U, X_R = x_R)\big] \leq \epsilon(1 - \delta),$$

*where $H[Z] = -\sum_{z \in [L]} \Pr(Z = z) \log \Pr(Z = z)$ is the entropy of the random variable $Z$.*

*Proof.* In this proof, we demonstrate the binary classification case. The extension to a multi-class scenario can be achieved through a similar process.

By the definition of the core feature set, there exists a representative label, denoted as $\tilde{y} \in \{0, 1\}$ such that the probability of $P(f_\theta(X_U, X_R = x_R) = \tilde{y})$ is greater than or equal to $1 - \delta$. Without loss of generality, we assume that the representative label is $\tilde{y} = 1$. Therefore, if we denote $Z$ as the probability of $Pr(f_\theta(X_U, X_R = x_R) = 1)$, then the probability of $Pr(f_\theta(X_U, X_R = x_R) = 0) = 1 - Z$. Additionally, we have $Z \geq 1 - \delta > 0.5$ due to the definition of core feature set and by the assumption that $\delta < 0.5$. The entropy of the model's prediction can be represented as: $H\big[f_\theta(X_U, X_R = x_R)\big] = -Z \log Z - (1 - Z) \log(1 - Z)$.

Choose $\epsilon(Z) = -Z \log Z - (1 - Z) \log(1 - Z)$. The derivative of $\epsilon(Z)$ is given by $\frac{d\epsilon(Z)}{dZ} = \log \frac{1-Z}{Z} < 0$, as $Z > 0.5$. As a result, $\epsilon(Z)$ is a monotonically decreasing function, so $\epsilon(Z) \leq \epsilon(1 - \delta)$

When $\delta = 0$, we have $Z = 1$, and by the property of the entropy $H\big[f_\theta(X_U, X_R = x_R)\big] = 0$. □

**Proposition 2.** *Given two subsets $R$ and $R'$ of sensitive features $S$, with $R \subseteq R'$,*

$$H\big(f_\theta(X_U, X_R = x_R)\big) \geq H\big(f_\theta(X_{U'}, X_{R'} = x_{R'})\big),$$

*where $U = S \setminus R$ and $U' = S \setminus R'$.*

*Proof.* This is due to the property that conditioning reduces the uncertainty, or the well-known *information never hurts* theorem in information theory [16]. □

**Proposition 3.** *The conditional distribution of any subset of unrevealed features $U' \in U$, given the the values of released features $X_R = x_R$ is given by:*

$$\Pr(X_{U'}|X_R = x_R) = \mathcal{N}\bigg(\mu_{U'} + \Sigma_{U',R}\Sigma_{RR}^{-1}(x_R - \mu_R), \ \Sigma_{U'U'} - \Sigma_{U'R}\Sigma_{RR}^{-1}\Sigma_{R,U'}\bigg),$$

*where $\Sigma$ is the covariance matrix*

*Proof.* This is a well-known property of the Gaussian distribution and we refer the reader to Chapter 2.3.2 of the textbook [8] for further details. □

**Proposition 4.** *The model predictions before thresholding, $\tilde{f}_\theta(X_U, X_R = x_R) = \theta_U X_U + \theta_R x_R$ is a random variable with a Gaussian distribution $\mathcal{N}\big(m_f, \sigma_f\big)$, where*

$$m_f = \theta_R x_R + \theta_U^\top\big(\mu_U + \Sigma_{UR}\Sigma_{RR}^{-1}(x_R - \mu_R)\big) \tag{8}$$

$$\sigma_f^2 = \theta_U^\top\big(\Sigma_{UU} - \Sigma_{UR}\Sigma_{RR}^{-1}\Sigma_{RU}\big)\theta_U, \tag{9}$$

*where $\theta_U$ is the sub-vector of parameters $\theta$ corresponding to the unrevealed features $U$.*

*Proof.* The proof of this statement is straightforward due to the property that a linear combination of Gaussian variables $X_U$ is also Gaussian. Additionally, the posterior distribution of $X_U$ is already provided in Proposition 3. □

**Proposition 5.** *Let the model predictions prior thresholding $\tilde{f}_\theta(X_U, X_R = x_R)$, be a random variable following a Gaussian distribution $\mathcal{N}(m_f, \sigma_f^2)$. Then, the model prediction following thresholding $f_\theta(X_U, X_R = x_R)$ is a random variable following a Bernoulli distribution $Bern(p)$ with $p = \Phi(\frac{m_f}{\sigma_f})$, where $\Phi(\cdot)$ refers to the CDF of the standard normal distribution, and $m_f$ and $\sigma_f$, are given in Equations (5) and (6), respectively.*

*Proof.* In the case of a binary classifier, we have $f_\theta(x) = \mathbf{1}\{\tilde{f}_\theta(x) \geq 0\}$. If $\tilde{f}$ follows a normal distribution, denoted as $\tilde{f} \sim \mathcal{N}(m_f, \sigma_f^2)$, then by the properties of the normal distribution, $f\theta$ follows a Bernoulli distribution, denoted as $f_\theta \sim Bern(p)$, with parameter $p = \Phi(\frac{m_f}{\sigma_f})$, where $\Phi(\cdot)$ is the cumulative density function of the standard normal distribution. $\qquad\square$

**Proposition 6.** *Assume $f_\theta$ is a linear classifier. Then, determining if a subset $U$ of sensitive features $S$ is a* pure *core feature set can be performed in $O(|P| + |S|)$ time.*

*Proof.* As discussed in the main text, to test if a subset $U$ is a core feature set or not, we need to check if the following two terms have the same sign (either negative or non-negative):

$$\max_{X_U} \theta_U^\top X_U + \theta_R^\top x_R = \|\theta_U\|_1 + \theta_R^\top x_R$$
$$\min_{X_U} \theta_U^\top X_U + \theta_R^\top x_R = -\|\theta_U\|_1 + \theta_R^\top x_R. \tag{10}$$

These can be solved in time $O(|P| + |S|)$ due to the property of the linear equality above. $\qquad\square$

**Theorem 1.** *The distribution of the random variable $\tilde{f}_\theta = \tilde{f}_\theta(X_U, X_R = x_R)$ where $X_U \sim \mathcal{N}(\mu_U^{pos}, \Sigma_U^{pos})$ can be approximated by a Normal distribution as*

$$\tilde{f}_\theta \sim \mathcal{N}\big(\tilde{f}_\theta(X_U = \mu_U^{pos}, X_R = x_R), g_U^\top \Sigma_U^{pos} g_U\big) \tag{11}$$

*where $g_U = \nabla_{X_U} \tilde{f}_\theta(X_U = \mu_U^{pos}, X_R = x_R)$ is the gradient of model prediction at $X_U = \mu_U^{pos}$.*

*Proof.* The proof relies on the first Taylor approximation of classifier $\tilde{f}$ around its mean:

$$\tilde{f}_\theta(X_U, X_R = x_R,) \approx \tilde{f}_\theta(X_U = \mu_U^{pos}, X_R = x_R) + (X_U - \mu_U^{pos})^T \nabla_{X_U} \tilde{f}_\theta(X_U = \mu_U^{pos}, X_R = x_R) \tag{12}$$

Since $X_U \sim \mathcal{N}(\mu_U^{pos}, \Sigma_U^{pos})$ hence $X_U - \mu_U^{pos} \sim \mathcal{N}(\mathbf{0}, \Sigma_U^{pos})$. By the properties of normal distribution, the right-hand side of Equation (12) is a linear combination of Gaussian variables, and it is also Gaussian. $\qquad\square$

# B    Algorithms Pseudocode

The pseudocode for MinDRel for non-linear classifiers is presented in Algorithm 2. There are two main differences between this algorithm and the case of linear classifiers. Firstly, unlike linear classifiers, the procedure of pure core feature testing on line 5 does not require the guarantee (see again Section 6.2). The accuracy of the testing procedures depends on the number of random samples that we evaluate. The greater the number of drawn samples, the more likely the testing procedure is to be accurate. During experiments, we draw $10^5$ samples to perform the testing. Additionally, we use Theorem 1 to estimate the distribution of the soft prediction as seen on line 11, as the exact distribution cannot be computed analytically as in the case of linear classifiers.

# C    Extension from binary to multiclass classification

In the main text, we provide the implementation of MinDRel for binary classification problems. In this section, we extend the method to the multiclass classification problem.

## C.1    Estimating $P(f_\theta(X_U, X_R = x_R))$

In order to achieve our goals of determining if a subset is a core feature set for a given $\delta > 0$, and computing the entropy in the scoring function, we need to estimate the distribution of $f_\theta(X_U, X_R = x_R)$. In this section, we first discuss the method of computing the distribution of $\tilde{f}_\theta(X_U, X_R = x_R)$ for both linear and non-linear models. Once this is done, we then address the challenge of estimating the hard label distribution $P(f\theta(X_U, X_R = x_R))$.

It is important to note that, under the assumption that the input features $X$ are normally distributed with mean $\mu$ and covariance matrix $\Sigma$, the linear classifier $\tilde{f}_\theta = \theta^\top x$ will also have a multivariate

**Algorithm 2:** MinDRel for non-linear classifiers

---

**input** : A test sample $x$; training data $D$
**output** : A core feature set $R$ and its representative label $\tilde{y}$

1  $\mu \leftarrow \frac{1}{|D|} \sum_{(x,y) \in D} x$
2  $\Sigma \leftarrow \frac{1}{|D|} \sum_{(x,y) \in D} (x - \mu)(x - \mu)^\top$
3  Initialize $R = \emptyset$
4  **while** *True* **do**
5      **if** *R is a core feature set with repr. label $\tilde{y}$* **then**
6          **return** $(R, \tilde{y})$
7      **else**
8          **foreach** $j \in U$ **do**
9              Compute $\Pr(X_j | X_R = x_R)$ (using Prop. 3)
10             $Z \leftarrow \text{sample}(\Pr(X_j | X_R = x_R))$ T times
11             Compute $\Pr\left(f_\theta(X_j = z, X_{U \setminus \{j\}} X_R = x_R)\right)$ using Theorem 1)
12             Compute $F(X_j)$ (using Eq. (4))
13     $j^* \leftarrow \text{argmax}_j F(X_j)$
14     $R \leftarrow R \cup \{j^*\}$
15     $U \leftarrow U \setminus \{j^*\}$

---

normal distribution. Specifically, if $X_U \sim \mathcal{N}(\mu_U^{pos}, \Sigma_U^{pos})$, then $\tilde{f}_\theta(X_U, X_R = x_R) \sim \mathcal{N}(\theta_R^\top x_R + \theta_U^T \mu_U^{pos}, \theta_U^\top \Sigma \theta_U)$.

For non-linear classifiers, the output $f_\theta(X_U, X_R = x_R)$ is not a Gaussian distribution due to the non-linear transformation. To approximate it, we use Theorem 1 which states that the non-linear function $\tilde{f}_\theta(X_U, X_R = x_R)$ can be approximated as a multivariate Gaussian distribution.

**Challenges when estimating** $P(f_\theta(X_U, X_R = x_R))$. For multi-class classification problems, the hard label $f_\theta(X_U, X_R = x_R)$ is obtained by selecting the class with the highest score, which is given by $\text{argmax}_{i \in [L]} \tilde{f}_\theta^i(X_U, X_R = x_R)$. However, due to the non-analytical nature of the argmax function, even when $\tilde{f}_\theta(X_U, X_R = x_R)$ follows a Gaussian distribution, the distribution of $f_\theta(X_U, X_R = x_R)$ cannot be computed analytically. To estimate this distribution, we resort to Monte Carlo sampling. Specifically, we draw a number of samples from $P(\tilde{f}_\theta(X_U, X_R = x_R))$, and for each class $y \in \mathcal{Y}$ we approximate the probability $P(f_\theta(X_U, X_R = x_R) = y)$ as the proportion of samples that fall in that class $y$.

We provide experiments of MinDRel for multi-class classification cases in Section D.5.

## D   Experiments details

**Datasets information.** To show the advantages of the suggested MinDRel technique for safe-guarding feature-level privacy, we employ benchmark datasets in our experiments. These datasets include both binary and multi-class classification datasets. The proposed method was evaluated on the following datasets for binary classification tasks:

1. Bank dataset [9]. The objective of this task is to predict whether a customer will subscribe to a term deposit using data from various features, including but not limited to call duration and age. There are a total of 16 features available for this analysis.

2. Adult income dataset [9]. The goal of this task is to predict whether an individual earns more than $50,000 annually. After preprocessing the data, there are a total of 40 features available for analysis, including but not limited to occupation, gender, race, and age.

3. Credit card default dataset [9]. The objective of this task is to predict whether a customer will default on a loan. The data used for this analysis includes 22 different features, such as the customer's age, marital status, and payment history.

4. Car insurance dataset [25]. The task at hand is to predict whether a customer has filed a claim with their car insurance company. The dataset for this analysis is provided by the insurance company and includes 16 features related to the customer, such as their gender, driving experience, age, and credit score.

Furthermore, we also evaluate the proposed method on two additional multi-class classification datasets:

1. Customer segmentation dataset [29]. The task at hand is to classify a customer into one of four distinct categories: A, B, C, and D. The dataset used for this task contains 9 different features, including profession, gender, and working experience, among others.
2. Children fetal health dataset [19]. The task at hand is to classify the health of a fetus into one of three categories: normal, suspect, or pathological, using data from CTG (cardiotocography) recordings. The data includes approximately 21 different features, such as heart rate and the number of uterine contractions.

**Settings.** For each dataset, 70% of the data will be used for training the classifiers, while the remaining 30% will be used for testing. The number of sensitive features, denoted as $|S|$, will be chosen randomly from the set of all features. The remaining features will be considered public. 100 repetition experiments will be performed for each choice of $|S|$, under different random seeds, and the results will be averaged. All methods that require Monte Carlo sampling will use 100 random samples. The performance of different methods will be evaluated based on accuracy and data leakage. Two different classifiers will be considered.

1. Linear classifiers: We use Logistic Regression as the base classifier.
2. Nonlinear classifiers: The nonlinear classifiers used in this study consist of a neural network with two hidden layers, using the ReLU activation function. The number of nodes in each hidden layer is set to 10. The network is trained using stochastic gradient descent (SGD) with a batch size of 32 and a learning rate of 0.001 for 300 epochs.

For Bayesian NN, we employ the package *bayesian-torch* [17] with the default settings. The base regressor is a neural network with one hidden layer that has 10 hidden nodes and a ReLU activation function. We train the network in 300 epochs with a learning rate of 0.001.

**Baseline models.** We compare our proposed algorithms with the following baseline models:

1. **All features**: This refers to the usage of the original classifier which asks users to reveal **all** sensitive features.
2. **Optimal**: This method involves evaluating all possible subsets of sensitive features ($2^{|S|}$ in total) in order to identify the minimum *pure* core feature set. For each subset, the verification algorithm is used to determine whether it is a pure core feature set. The minimum pure core feature set that is found is then selected. It should be noted that as all possible subsets are evaluated, all sensitive feature values must be revealed. Therefore, this approach is not practical in real-world scenarios. However, it does provide a lower bound on data leakage for MinDRel (when $\delta = 0$).

**MinDRel models.** In MinDRel there are two important steps: (1) core feature set verification and (2) selection next feature to reveal. As additional baselines, we keep the core feature set verification and vary the selection process. We consider the following three feature selection methods:

1. **F-Score**: We choose the feature based on the amount of information on model prediction we gain after revealing one feature as provided in Equation 3.
2. **Importance**: We reveal the unknown sensitive features based on the descending order of feature importance until we find a core feature set. The feature importance is determined as follows. We firstly fit a Logistic Regression $f_\theta(x) = 1\{\theta^T x \geq 0\}$ on the training dataset $D$ using all features (public included). The importance of one sensitive feature $i \in S$ is determined by $\|\theta_i\|_2$.
3. **Random**: We reveal the unrevealed sensitive feature in random order until the revealed set is a core feature set.

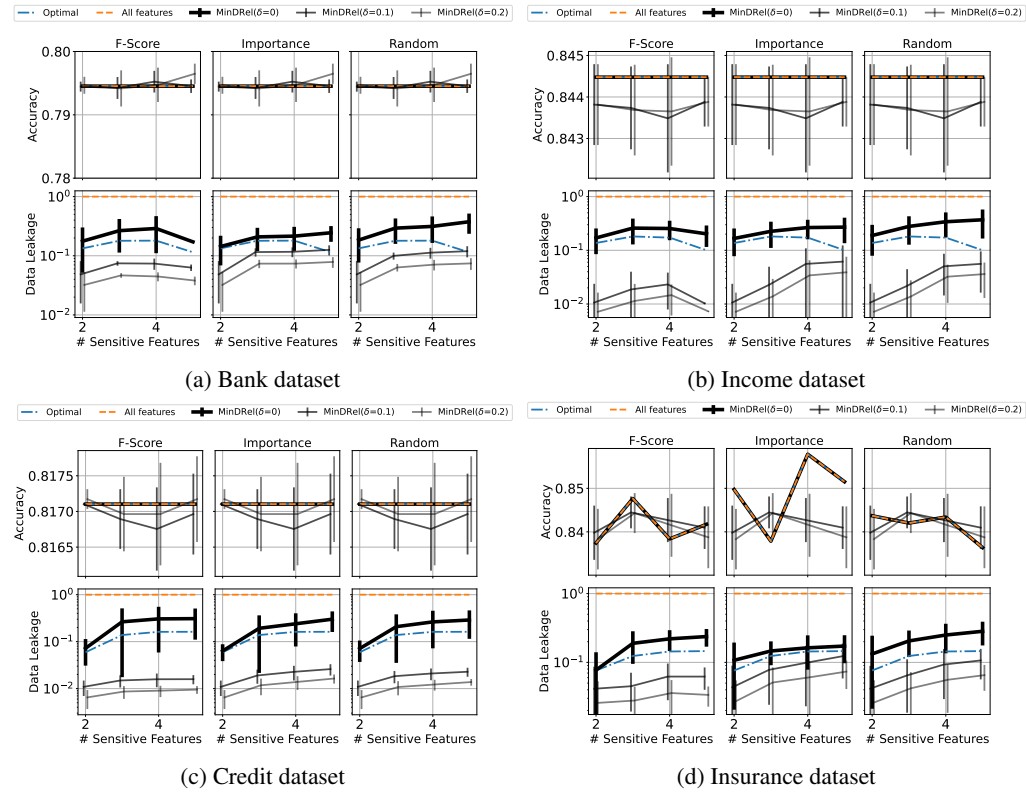

(a) Bank dataset

(b) Income dataset

(c) Credit dataset

(d) Insurance dataset

Figure 7: Comparison between using (left) our proposed F-Score (left) with Importance (Middle) and Random (Right) for different choices of the number of sensitive features $|S|$. The baseline classifier is Logistic Regression

**Metrics.** We compare all different algorithms in terms of accuracy and data leakage:

1. Accuracy. For algorithms that are based on the core feature set, such as our MinDRel and Optimal, the representative label is used as the model's prediction. Again, the representative label for $\delta = 0$ can be identified by using testing pure core feature set procedures. For $\delta > 0$, the representative label is given by $\tilde{y} = \text{argmax}_{y \in \mathcal{Y}} \int P(f_\theta(X_U = x_U, X_R = x_R) = y)P(X_U|X_R = x_R)dx_u$. The accuracy is then determined by comparing this representative label to the ground truth.

2. Data leakage. We compute the percentage of the number of sensitive features that users need to provide on the test set. Small data leakage is considered better.

### D.1 Additional comparison between using Gaussian assumption and Bayesian NN

We first show empirically the benefits of our proposed Gaussian assumption compared to using Bayesian NN which allows more flexibility in modeling the conditional distribution $P(X_U|X_R = x_R)$. We report both training and inference time between Bayesian NN and our Gaussian assumption on various datasets when the number of sensitive features $|S| = 5$ in Table 1 and Table 2. When $|S| = 5$ the number of possible subsets $U \in S$ is $2^5 = 32$ which requires training 32 Bayesian NN models. This will be especially slow for datasets with a large number of training samples (e.g., Income with 50K samples). In contrast, using Gaussian assumption we just need to precompute 32 inverse matrices $\Sigma_{R,R}^{-1}$ which is pretty fast for data that have a small number of features (less than 50 in our experiments). It is noted again that in this paper we focus on the case when the number of training samples is much more than the number of features. Likewise, during inference time, with Gaussian assumption, we can compute the distribution of model prediction in a closed form by simple matrix multiplication which takes $O(d^2)$. Instead, using Bayesian NN, it requires expensive Monte Carlo sampling, especially when $|U|$ is large to obtain an accurate estimation of $P(X_U|X_R = x_R)$.

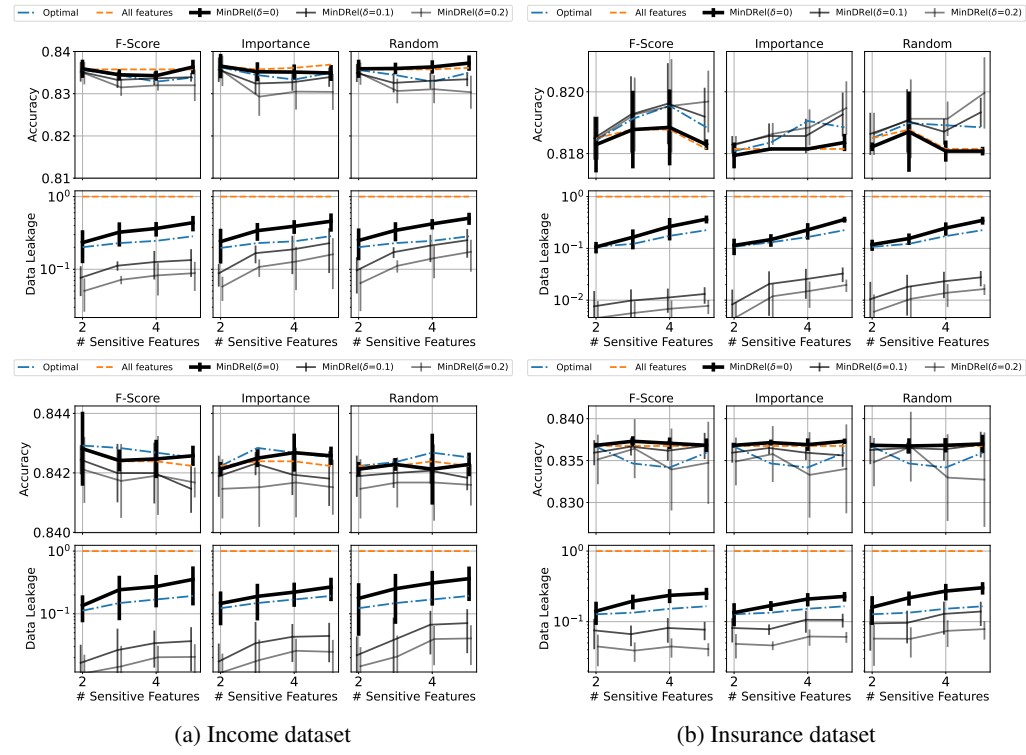

| (a) Income dataset | (b) Insurance dataset |

Figure 8: Comparison between using (left) our proposed F-Score (left) with Importance (Middle) and Random (Right) for different choices of the number of sensitive features $|S|$. The baseline classifier is a neural network classifier.

| Method | Bank | Income | Credit | Insurance |
|---|---|---|---|---|
| Bayesian NN | 204 | 375 | 125 | 90 |
| Gaussian assumption | 0.01 | 0.02 | 0.02 | 0.01 |

Table 1: Comparison between using Bayesian neural network and our Gaussian assumption in terms of training time (minutes) when |S| = 5 for various datasets.

We also report the performance in terms of accuracy and data leakage between using Gaussian assumption and Bayesian NN in Figure 9. We see no significant difference in terms of accuracy and data leakage between the two choices of modeling $P(X_U|X_R = x_R)$. In addition, as indicated above using the Gaussian assumption reduces significantly the training and inference time, in the subsequent experiments we will use the Gaussian assumption in MinRDel with F-Score selection.

## D.2 Additional experiments on linear binary classifiers

Additional experiments were conducted to compare the performance of MinDRel to that of the baseline methods using linear classifiers on the Bank, Adult income, Credit, and Insurance datasets, as shown in Figure 7. As in the main text, a consistent trend in terms of performance is observed.

| Method | Bank | Income | Credit | Insurance |
|---|---|---|---|---|
| Bayesian NN | 40 | 254 | 220 | 34 |
| Gaussian assumption | 15 | 78 | 66 | 9 |

Table 2: Comparison between using Bayesian neural network and our Gaussian assumption in terms of inference time (minutes) on the test set when $|S| = 5, \delta = 0$ for various datasets.

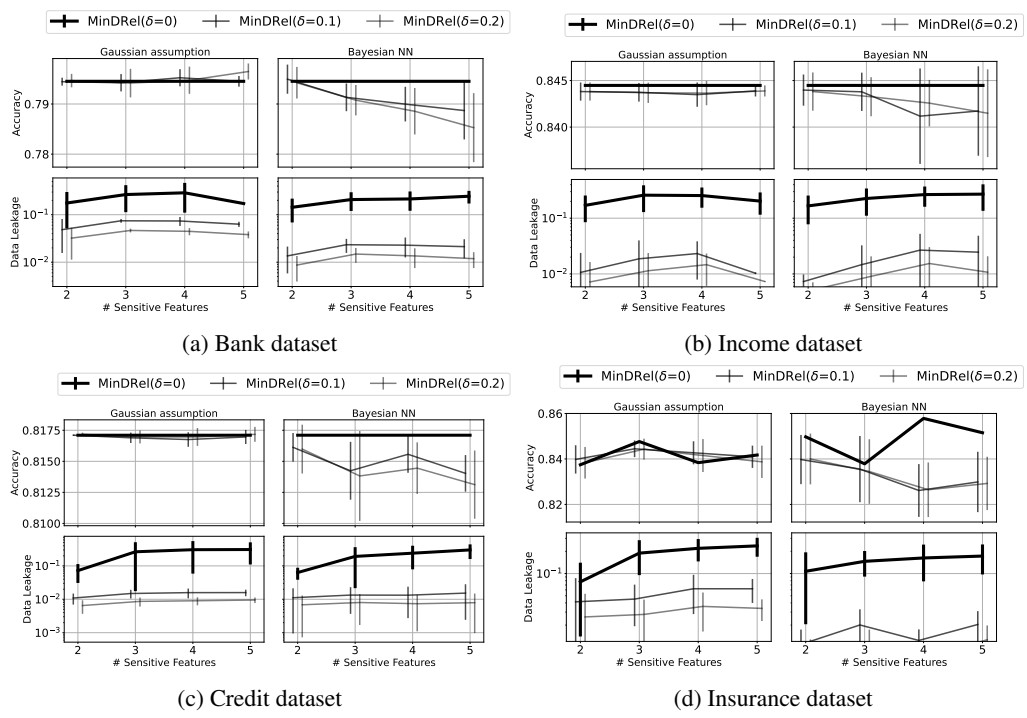

Figure 9: Comparison between using Bayesian NN with our Gaussian assumption in terms of (1): accuracy and (2) data leakage for different choices of the number of sensitive features $|S|$ on different datasets using a Logistic Regression classifier.

As the number of sensitive attributes, $|S|$, increases, the data leakage introduced by MinDRel with various values of $\delta$ increases at a slower rate. With different choices of $|S|$, MinDRel (with $\delta = 0$) requires the revelation of at most 50% of sensitive information. To significantly reduce the data leakage of MinDRel, the value of $\delta$ can be relaxed. As mentioned in the main text, $\delta$ controls the trade-off between accuracy and data leakage here. The larger $\delta$ is, the greater uncertainty the model prediction has, which implies the fewer sensitive features users need to reveal and the lower accuracy of the model prediction. By choosing an appropriate value for the failure probability, such as $\delta = 0.1$, only minimal accuracy is sacrificed (at most 0.002%), while the data leakage can be reduced to as low as 5% of the total number of sensitive attributes.

### D.3 Additional experiments on non-linear binary classifiers

Additional experiments were conducted to compare the performance of MinDRel to that of the baseline methods using non-linear classifiers on the Bank, Adult income, Credit, and Insurance datasets, as shown in Figure 8. As seen, while the baseline **All features** method requires the revelation of all sensitive attributes, MinDRel with different values of $\delta$ only requires the revelation of a much smaller number of sensitive attributes. The accuracy difference between the Baseline method and MinDRel is also minimal (at most 2%). These results demonstrate the effectiveness of MinDRel in protecting privacy while maintaining a good prediction performance for test data.

### D.4 Sclability of MinDRel for large $|S|$

We demonstrate the performance of MinDRel when we have a large number of sensitive features $|S|$. Note that to reduce the runtime we did not run the *Optimal* method which performs an exponential search over all possible choices of the subset of $S$.

We first report the accuracy and data leakage of MinDRel when using F-Score or using either two heuristic rules Importance and Random in the case of logistic regression classifiers in Figure 10.

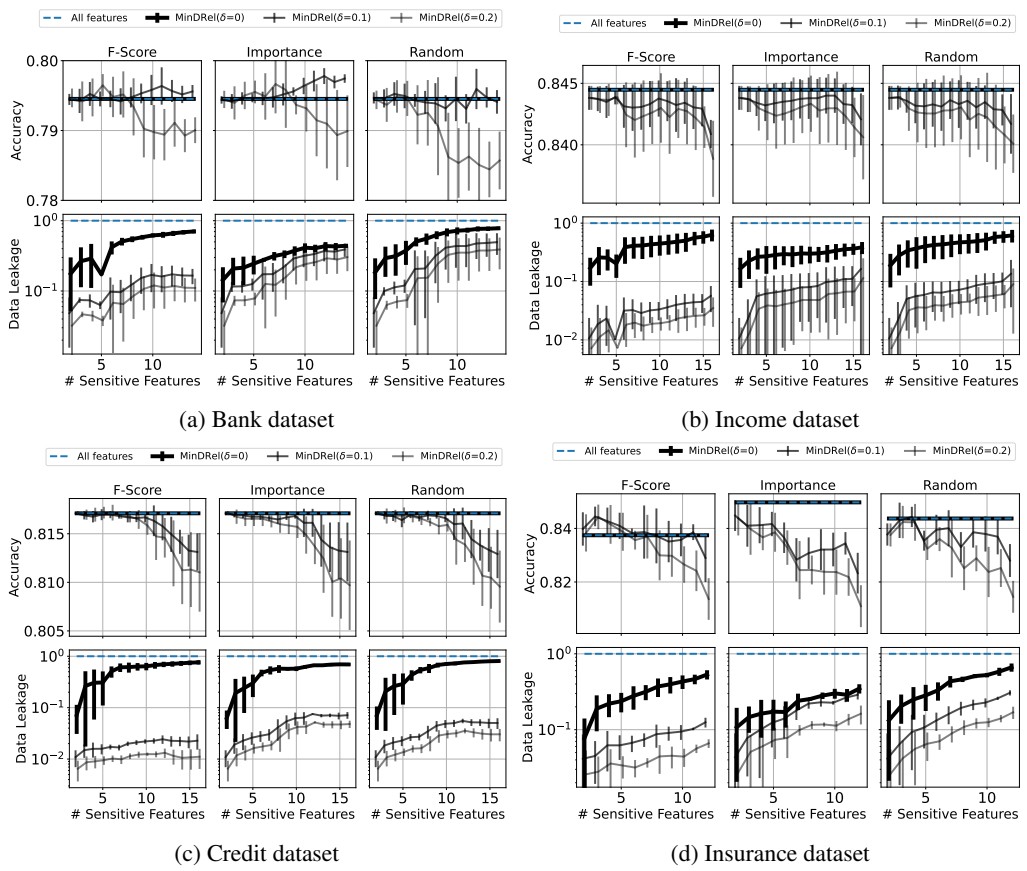

(a) Bank dataset        (b) Income dataset

(c) Credit dataset        (d) Insurance dataset

Figure 10: Comparison between using (left) our proposed F-Score (left) with Importance (Middle) and Random (Right) for different choices of number of sensitive features $|S|$. The baseline classifier is a logistic regression classifier.

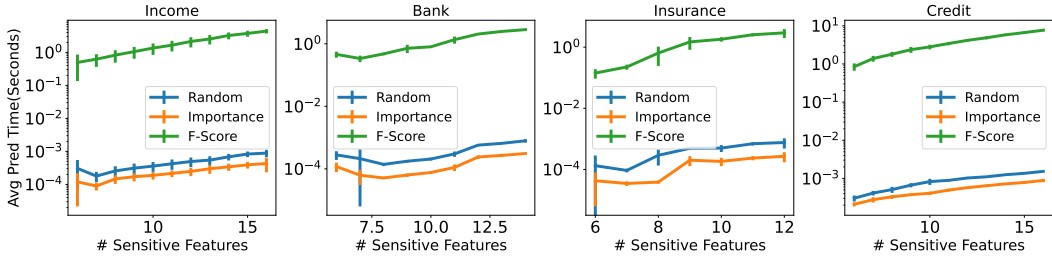

Figure 11: Comparison in terms of average prediction time (seconds) among F-Score, Importance and Random method in MinDRel ($\delta = 0$) for different $|S|$.

Finally, we report the average testing time (in seconds) to get the model prediction per user of MinDRel in Figure 11. It is noted that in this case, we assume the time taken by users to release sensitive features is negligible. It is evident that when $|S| > 15$, our proposed MinDRel with F-Score can take slightly more than 1 second to get the model prediction per user. This demonstrates the applicability of the models in practice.

### D.5 Evaluation of MinDRel on multi-class classifiers

**Linear classifiers**    We also provide a comparison of accuracy and data leakage between our proposed MinDRel and the baseline models for linear classifiers. These metrics are reported for the Customer and Children Fetal Health datasets in Figures 12a and 12b, respectively. The figures clearly show

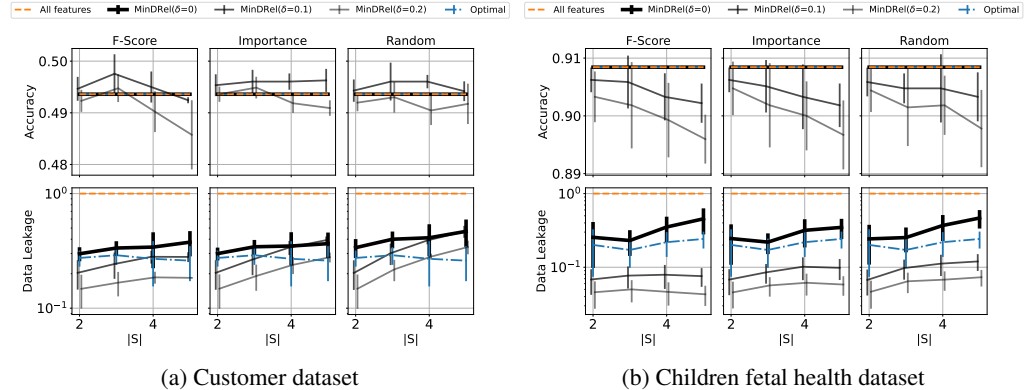

(a) Customer dataset                (b) Children fetal health dataset

Figure 12: Comparison between using our proposed F-Score (left) with Importance (Middle) and Random (Right) for different choices of the number of sensitive features $|S|$. The baseline classifier is a multinomial Logistic Regression.

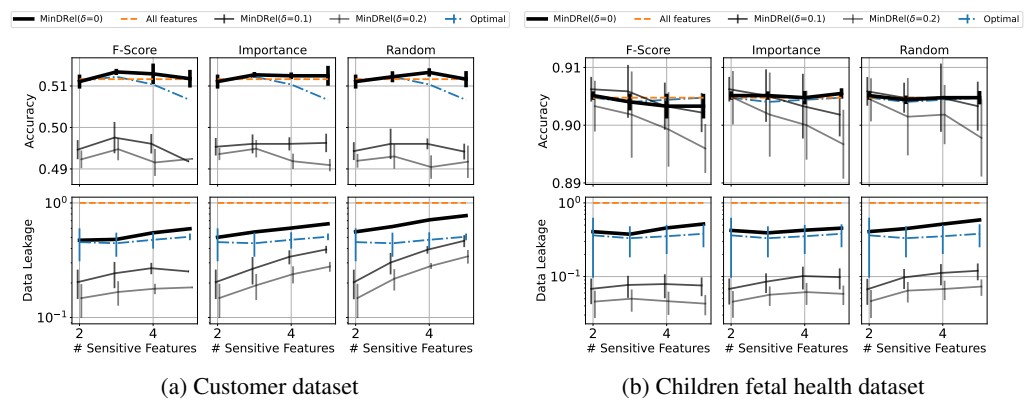

(a) Customer dataset                (b) Children fetal health dataset

Figure 13: Comparison between using our proposed F-Score (left) with Importance (Middle) and Random (Right) for different choices of the number of sensitive features $|S|$. The baseline classifier is a neural network classifier.

the benefits of MinDRel in reducing data leakage while maintaining a comparable accuracy to the baseline models.

**Nonlinear classifiers**    Similarly, we present a comparison of our proposed algorithms with the baseline methods when using non-linear classifiers. These metrics are reported for the Customer and Children Fetal Health datasets in Figures 13a and 13b, respectively. The results show that using MinDRel with a value of $\delta = 0$ results in a minimal decrease in accuracy, but significantly reduces the amount of data leakage compared to the Baseline method.

## E    Exetended Motivation

Finally, we delve deeper into the motivational aspects underscoring our research. Our study primarily concerns tabular datasets where the total number of features is significantly outnumbered by the training samples. We concentrate on applications necessitating user-provided personal information for decision-making, such as lending, online insurance services, and healthcare services. The prevailing weaknesses of these applications, which our work attempts to address, are outlined as follows:

1. Privacy Concerns: The primary issue stems from the need for users to disclose sensitive information. For instance, in the context of online healthcare services, patients are required to share an array of sensitive health data—weight, height, smoking habits, etc.—via a website or mobile application. This exposes users to potential privacy threats.

2. User and Organizational Expenditure of Time and Effort: Numerous applications, such as lending, involve the time-consuming and effort-intensive process of gathering sensitive information and its supporting evidence. Users, for example, are required to validate their income through payslips or employment contracts in lending applications. Similarly, the organization must invest time and resources to verify the authenticity of submitted documents.

3. Legal Constraints: According to the EU General Data Protection Regulation's principle of data minimization, the collection of excessive personal information by companies and organizations is restricted. Our primary text illustrates that it is not imperative to report all features to preserve the model's accuracy.

**Inference Time.** We also emphasize the importance of low inference time from both user and business perspectives. From the user's viewpoint, applications such as online car insurance require answering a series of questions to determine the insurance plan. Naturally, users prefer answering fewer questions in the least amount of time. From a business standpoint, prolonged inference time may lead to customer dissatisfaction, potentially resulting in contract termination. This serves as the basis for our algorithmic choices, in lieu of more complex conditional distribution modeling methods, which can significantly increase inference time.

