# OpenReview forum: "Data Minimization at Inference Time"
_NeurIPS.cc/2023/Conference — NeurIPS 2023 poster_

### Official Review · Reviewer_ZsPJ · 2023-07-06

**Soundness:** 3 good
**Presentation:** 2 fair
**Contribution:** 4 excellent
**Rating:** 6
**Confidence:** 4

**Summary:**

This paper questions the necessity of soliciting all user information at inference time, with an eye towards privacy-sensitive domains such as law, recruitment, and healthcare, where learning models typically require access to extensive and sensitive user data for accurate prediction. The authors propose in some settings, individuals might only need to provide a small subset of their information to obtain accurate predictions, thereby protecting their privacy and easing the burden on organizations verifying the accuracy of disclosed information. To that end, the authors introduce a framework for simultaneously considering model certainty and data minimization. Their setup includes considering subsets of features that individuals can disclose in order to receive comparable prediction quality while providing fewer features. The authors propose algorithms for choosing the appropriate subset and present theoretical justification and empirical findings. Initial evaluations reveal that individuals may only need to disclose about 10% of their information to achieve the same accuracy level as a model that uses all user information.

**Strengths:**

The primary strength of this paper is that it thoughtfully addresses a neglected and high-impact research question. As the authors note, there is strong evidence that "most users would only need to reveal a small portion of their sensitive data to achieve accurate model predictions with either absolute certainty or high confidence." They include a strong motivating example in Section 2, Figure 1, to make it clear that it's possible that a label could be obtained without requesting all sensitive features. Similarly, they underscore the importance of this work by noting the real-world impacts on privacy and the burden of verifying the provided information.

Other strengths:
- This paper includes a thoughtful framework for studying this problem, quantifying the tradeoff between model certainty (though not model performance, as claimed) and privacy loss.
- The proposed methods include an efficient algorithm for a joint Gaussian setting as well as a less efficient Bayesian modeling alternative
- The experiments appear well-designed, including considering random subsets of "private" features using publicly available datasets

**Weaknesses:**

The primary weakness of this work is the use of entropy as a proxy for model performance. While the authors state that initially that their goal is the "produce accurate or nearly accurate predictions during inference" using fewer features, the goalposts subtly shift to "accurately predict[ing] the **output of the model** using the smallest possible number of sensitive features." This shift motivates the use of prediction entropy, rather than predictive performance, as a metric for the impact of obtaining additional features. This is significant because while entropy obeys the "information never hurts" principle, predictive performance does not (see, e.g., https://proceedings.mlr.press/v97/ustun19a.html). Considering predictive performance (instead of entropy) likely makes this problem more difficult, and it's possible that using entropy is a reasonable proxy in some cases. However, this choice merits at minimum discussion and ideally theoretical and experimental justification.

Other weaknesses:

Unclear/imprecise notation and language:
- "data leakage" is used inconsistently throughout the paper. It is introduced as "the percentage of sensitive features that are revealed unnecessarily, meaning that their exclusion would not significantly impact the model’s output," but the authors later state that "[increasing $\delta$ yields] reduced data leakage but also less precise model predictions" which appears contradictory. This term is later used as a metric in the experiments, but without a precise definition and with axes that seem to contradict prior use.
- on line 155-156, the authors state that the imputation does not occur, and the "unrevealed features are treated as random variables and are integrated during the prediction process." This process is not further explained but appears to be implicit in Equation 5 with a Schur complement (and arguably as a form of imputation).
- "public" is used both to describe a subset of features "public x_P and sensitive x_S features" and a subset of samples "trained on a public dataset"
- The hyperparameter T appears on line 9 of Algorithm 1 but is not discussed from what I could tell, either theoretically or empirically
- In line 13 of Algorithm 1, it is unclear if feature j* is actually obtained here or if it is added to the set of features to be obtained. I assume the former.
- The assumption that the cost of obtaining each feature is uniform should be explicitly stated.


Other:
- The authors claim throughout the paper that they are the first to study this type of work (line 47, line 420) despite the presence of several other publications on this topic (e.g., https://arxiv.org/abs/1602.03600 or https://proceedings.mlr.press/v5/yu09a.html)
- Some words were used improperly (e.g., "contrast" on line 9, "recur" on 196, "valid" (sufficient?) on line 227). Overall, could use another pass to make the writing more clear.

**Questions:**

- Can you explain or justify your use of model entropy as a proxy for model performance?
- The thresholding explanation on line 190 was a bit confusing -- can you explain this process in more detail?

**Limitations:**

Yes

---

> ### Author Rebuttal · Authors · 2023-08-08
>
> Thank you for your time and review. Below, we report our answers to your questions. Feel free to let us know if there are further questions or concerns and we'll be more than happy to elaborate.
>
>
> ### Comment on hyperparameter T
>
> You are right, and we appreciate the catch. $T$ is indeed the number of Monte Carlo samples used in the algorithm. We acknowledge that this was not clearly explained in the original text, and we will include a more detailed explanation of this hyperparameter in our revision.
>
> ### Comment on line 13 of Algorithm 1
>
> Your assumption is correct. In line 13 of Algorithm 1, we identify the most promising unrevealed feature $j^\star$ that provides the most information about the model's prediction. The value of this feature is then obtained from the user, and we remove it from the set of unrevealed features $U$ and add it to the set of revealed features
> $R$.
>
> ### Comment on cost of obtaining each feature
>
> Thank you for highlighting this point. We indeed operate under the scenario in which the cost, whether it's the privacy cost or the cost of obtaining the feature value for sensitive features, is uniform across the features. This assumption will be explicitly stated in our revised manuscript to ensure clarity.
>
> ### Comment on related work
>
> We appreciate the reviewer for highlighting these relevant works. Upon review, we found that the papers shared focus on different aspects, such as training time, reinforcement learning (in the first paper), and multi-view learning (in the second paper).
>
> Our work, in contrast, specifically targets the testing phase where a pretrained classifier is given. Therefore, while the cited works are indeed valuable and related, the context and focus of our research distinguish it from the prior studies. This helps clarify the claim made in lines 47 and 420 about the novelty of our approach in this particular domain.
>
> We will be happy to discuss such related work in the final version of our paper (please also see current discussion of related work in Appendix B).
>
> ### Q1: Model performance
>
> Indeed, the use of model entropy in our work serves as a tool to measure the uncertainty associated with the model's predictions. We focus on the trade-off between revealing sensitive features and maintaining prediction accuracy and want to achieve better accuracy for the whole population while minimizing data leakage, which includes revealing fewer features.
>
> The model entropy helps us quantify the uncertainty in the model's predictions. As we reveal more features, the entropy typically decreases, reflecting that the model becomes more confident in its predictions. Conversely, if we reveal fewer features, the entropy may increase, indicating higher uncertainty.
>
> However, it is important to note that revealing more sensitive features does not necessarily imply higher prediction accuracy for some individuals. This is obviously important in personalization settings.
> However, and importantly, in our study, we have empirically observed that revealing more features generally ensures better accuracy across the whole population (see, for instance Figure 6). But designing a personalized algorithm that considers both data leakage and accuracy can be challenging, and this relationship may not hold for every individual.
>
> ### Q2: Further explanation of line 190
>
> Absolutely! In our framework, we have to compute the distribution of $\tilde{f}_\theta$ over the uncertainty of certain unrevealed sensitive features. If the distribution of these unrevealed features follows a Gaussian distribution, then the distribution of the soft-label prediction (i.e., $f_\theta(x) = \theta^\top x$) is also linear, or approximately linear in the case of non-linearity.
>
> Now, here's where thresholding comes into play. To move from the soft-label prediction to a hard-label prediction, we apply a thresholding operation. Since the soft-label prediction follows a Gaussian distribution, thresholding this Gaussian turns it into a Bernoulli variable. In other words, we are converting a continuous-valued prediction into a binary decision by comparing it to a threshold.
>
> The thresholding process is a key step in translating continuous predictions into categorical outcomes, especially when we are dealing with uncertainties tied to unrevealed features. We understand that the concise description in line 190 might have been unclear, and we plan to include a more extended and illustrative description in the final version of the paper to alleviate any confusion.
>
> We hope we have addressed all your concerns, and we welcome any further questions or feedback.

---

> > ### Author Response · Authors · 2023-08-13
> >
> > We wanted to reiterate our gratitude for your time and review and would like to check if you had any additional questions or comments.
> >
> > Also, it appears we had a formatting issue in the previous response. It should read:
> >
> > Absolutely! In our framework, we have to compute the distribution of $\tilde{f}_{\theta}$  over the uncertainty of certain unrevealed sensitive features.
> > If the distribution of these unrevealed features follows a Gaussian distribution, then the distribution of the soft-label prediction is also linear, or approximately linear, in the case of non-linearity.
> >
> > Many thanks!

---

> > ### Comment · Reviewer_ZsPJ · 2023-08-17
> >
> > I appreciate your clarifications and think adding the revisions you've described here will improve the manuscript nicely.
> >
> > Regarding the choice of entropy as a metric, I still find your explanation and justification unsatisfying. If this work aims to "achieve better accuracy for the whole population while minimizing data leakage," then using model certainty or entropy as a proxy for accuracy requires theoretical and/or experimental justification. While you noted general relationships between disclosure and entropy and between disclosure and performance, these trends do not lead to the conclusion that entropy is a sufficient proxy for performance. While as you noted, exploring the relationship between disclosure and performance is difficult, one alternative is to make it clear that this paper focuses on improving **certainty** (not accuracy/performance) while minimizing "data leakage," and to remove claims about improving performance.

---

> > > ### Author Response · Authors · 2023-08-18
> > >
> > > Thank you for your comment. We agree with your suggestion and will make it clearer in the paper that our primary focus is on minimizing data leakage while improving certainty, and, at the same time, ensuring that our claims are consistent with our focus and the evidence provided.
> > >
> > > Note also that in our work, entropy is used as a link to core feature sets, which, in turn, captures our concept of data leakage, which is of central interest in data minimization.
> > >
> > > Providing certificates on accuracy would indeed be a useful desiderata and is a topic of future exploration, but, we also note that this is the first work exploring data minimization at inference time and we believe will pave the way to additional significant contributions. Thank you again, we appreciate your support!

---

### Official Review · Reviewer_ddiE · 2023-07-08

**Soundness:** 2 fair
**Presentation:** 2 fair
**Contribution:** 2 fair
**Rating:** 5
**Confidence:** 4

**Summary:**

The authors address the problem of data minimization at inference time, which poses a real-world challenge in real-world applications, where users might want to hide sensitive or personal attributes. They provide an efficient algorithm to sequentially determine the appropriate attributes for an individual, with the goal to maintain the original predictive accuracy (based on all attributes).

**Strengths:**

1. The considered problem is important and timely.
2. The proposed theoretical framework introduces some interesting novel concepts.
3. The paper is well motivated and well-written.

**Weaknesses:**

1. The addressed problem is inherently a privacy problem; if some feature values are not explicitly available at inference time, it does not mean that they can not be inferred. Actually, in practice if the predictive quality does not change after hiding some feature values, it is a strong indication that these values can be inferred from the other values.

2. The overall objective of maintaining the original predictive quality with the minimum amount of private features is never explicitly formalized/defined.

3. In many real-world applications that use personal data, accuracy is the worst measure for predictive quality, so focusing on and reporting this measure is not really meaningful (e.g., when the goal is good model calibration on imbalanced classes).

4. Relevant related work has not been considered: I Prefer not to Say: Operationalizing Fair and User-guided Data Minimization
T Leemann, M Pawelczyk, CT Eberle, G Kasneci

**Questions:**

Can you comment on the weakness above and limitations below?

**Limitations:**

The novelty of the work is rather limited, as relevant work on this topic has already addressed similar challenges.
The formalization of the approach and the theoretical framework is limited as it mainly aims for maintaining the high accuracy, and it does not consider the true privacy of the features that users wish to hide.
The evaluation is also focused on accuracy, which is not a meaningful measure in real-world ML applications based on tabular user data.

---

> ### Author Rebuttal · Authors · 2023-08-08
>
> Thank you for your time and review. Below, we report our answers to your questions. Feel free to let us know if there are further questions or concerns and we'll be more than happy to elaborate.
>
> ### Comment on `The addressed problem is inherently a privacy problem ... can be inferred from the other values`
>
> We appreciate your perspective on the inherent privacy concerns in the problem we've addressed. However, we'd like to clarify that the situation you describe might not necessarily hold in our context.
>
> To illustrate our point, let's revisit the motivating example from Section 2 of our paper. In this example, the model's prediction remains invariant with respect to sensitive attributes like _Income_ and _Location_ when _Job_ equals 1. This behavior holds true even if all variables are independent.
>
> Thus, in this particular case, the fact that the predictive quality does not change after hiding some feature values does not imply that these values can be inferred from the other values. Given the published _Job_ information, it is ineffective to use the model predictions to infer the original values of _Income_ and _Location_.
>
> We understand the broader concern about potential inference from hidden features, and we agree that it's a critical consideration in privacy-related research. However, the specific mechanism we've explored offers safeguards against such inference in the scenarios we've examined.
>
> We're open to further discussion if you have additional concerns or insights.
>
> ### Q2 formalization of our objective
>
> we have indeed described the overall objective of maintaining the original predictive quality with the minimum amount of private features. This description can be found in Line 65 of Section 2.
> We chose to express this goal through an English description, believing it to be sufficiently clear and avoiding additional mathematical notation that might complicate the reading.
>
> However, we are receptive to your feedback and we are certainly open to incorporate a formal definition upon paper acceptance.
>
>
> ### Q3 Choice of accuracy metric
>
> We acknowledge that accuracy may not be suitable for all real-world applications, however, in the context of our work, we adopted accuracy as the evaluation metric for specific reasons.
>
> - First, we considered both binary and multiclass classification problems, making AUC less applicable in our situation. This decision is elaborated further in the Appendix.
>
> - Second, and more crucially, our work emphasizes obtaining the model's hard prediction, where confidence can be assessed with 100% certainty even when some sensitive features remain unrevealed. This scenario differs fundamentally from soft-label prediction, where a full confidence estimation of the prediction score is impossible if any features are undisclosed.
>
> In essence, our focus on hard-label prediction, as opposed to soft-label prediction, requires users to reveal less sensitive information. This approach is aligned with the concept of minimum uncertainty, allowing us to achieve the highest confidence in our predictions.
>
> We of course are open to considering other suitable metrics and appreciate your insights, but we also hope this explanation clarifies our choice of accuracy as the evaluation metric.
>
>
> ### Q4 related work
>
> Thank you for bringing the paper by Leemann et al. to our attention. We are aware of this work. It focuses on the privacy concerns related to opting out from data collection for individuals who choose not to share certain information.
> While the subject of privacy is a common theme between our work and the paper you mentioned, the direction and approach taken in our research are quite distinct. In our study, we are primarily concerned with minimizing data leakage by revealing the least amount of sensitive features without affecting prediction accuracy.
>
> Although the connection between the two works is somewhat tangential, we will consider adding a reference to this paper in the final version of our work for completeness and to acknowledge the broader context of privacy research.
>
>
>
> We appreciate your feedback and hope you could reconsider your score, in light of our responses to your questions.

---

> > ### Author Response · Authors · 2023-08-13
> >
> > We wanted to reiterate our gratitude for your time and review and would like to check if you had any additional questions or comments. Many thanks!

---

> > ### Comment · Reviewer_ddiE · 2023-08-13
> > **Thank you for your reply!**
> >
> > Thank you for your reply. Please find my answers below.
> > Q1: “it is ineffective to use the model predictions to infer the original values of Income and Location“ ineffective doesn’t mean impossible. If it is possible to infer the original values of the sensitive values, your approach misses the motivational point, because what is the point of identifying the “appropriate attributes for each individual to provide“ if it is not privacy?
> >
> > Q2: A mathematical formulation of the objective would certainly be less ambiguous.
> >
> > Q3: I find it quite impractical to predict the hard labels directly. But if you decide to do so, you can still conduct a precision-recall analysis (in addition to accuracy). For multi-class cases, you can consider micro and macro averages.
> >
> > Q4: I think broadening the spectrum of different contexts (and related approaches) in which users would like to hide or provide sensitive information would greatly benefit the current work.
> >
> > For now, I will keep my initial score.

---

> > > ### Author Response · Authors · 2023-08-14
> > >
> > > Thanks for answering our rebuttal.
> > >
> > > **Q1**: We urge you to review again the motivating example in Figure 1. We believe this will clarify your question. In that example, it is impossible to infer the sensitive attributes (location, Income) when Job, Loc, and Inc are mutually independent, i.e., $\Pr(\text{Loc}, \text{Inc}) = \Pr(\text{Loc}, \text{Inc} | \text{Job})$.
> > >
> > > However, if we observe $\text{Job} =1$, we know for sure $1* \text{Job} - 0.5 * \text{Loc} + 0.5 * \text{Inc} >= 1*1 - 0.5 * \text{Loc} + 0.5 * \text{Inc} >=0$ for any arbitrary values of Loc, Inc. Hence, we know the model prediction (**not the ground truth**) of that individual even though have not observed their Inc and Loc feature values.
> > >
> > > We notice that this reviewer has mentioned earlier the paper "I Prefer not to Say: Operationalizing Fair and User-guided Data Minimization T Leemann, M Pawelczyk, CT Eberle, G Kasneci". Our setting, motivation, and privacy notion are certainly different from those in such a paper.
> > > In our context, users are not given the option to choose which features to release; instead, entities such as the system, the bank, or the insurance company make that decision. The term "appropriate attributes" refers to specific features that the system believes, if released, can provide the most insight or additional information regarding the model's predictions. When all features are independent, the model will of course select the most important one. However, note that we focus on minimizing data at inference time, and the determination of the "most important" features is based on the training data, which is publicly available in our setting. Therefore, there is no privacy loss incurred in this process, within the setting of our paper.
> > >
> > > We will certainly take your other points into consideration. Thank you for the feedback!
> > >
> > > From your response, it also appears that we may have clarified all your concerns. Let us know if this is not the case and again we hope you could reconsider your score, in light of our responses to your questions.

---

> > > > ### Comment · Reviewer_ddiE · 2023-08-14
> > > > **I think you are imposing too many unrealistic constraints**
> > > >
> > > > Thank you again for your reply. Despite the fact that the motivating example is quite unrealistic, I agree that for independent features, the privacy desideratum for the sensitive attributes holds. Nevertheless, the independence assumption and the prediction of hard labels (which builds the basis of your approach) are quite impractical. I know the banking and financial services sector very well, and please believe me that the probability of default (PD) is the most fundamental concept/tool for the processes in this sector. Even when hard labels are used (e.g., for rule-based decision making in this sector), the PD ranges behind the labels are very well known and model calibration is a prime concern.  Hence, I am not convinced by the usability aspect of the approach.

---

> > > > > ### Author Response · Authors · 2023-08-14
> > > > >
> > > > > We want to make sure we convey the message that **our work does not rely on such an independence assumption**. We hope this is clear.

---

> > > > > > ### Author Response · Authors · 2023-08-18
> > > > > > **Summary of discussion**
> > > > > >
> > > > > > Thank you for the insightful discussion thus far. We'd like to provide a summary of our conversation and provide additional clarification, which, we hope serves to clarify all your doubts.
> > > > > >
> > > > > > **1.** Your initial question concerned the ability to reconstruct attributes that haven't been released.
> > > > > >
> > > > > > In many cases, reconstructing non-released attributes with high confidence (or 0-error) relies on models trained on redundant or highly correlated features. However, our work is about data minimization at test time and focuses on datasets lacking such redundancy.
> > > > > > This is indeed the case in practice, where training datasets are curated to exclude highly redundant features.
> > > > > >
> > > > > > To provide further evidence, we've conducted new experiments to demonstrate the ineffectiveness of classical reconstruction attacks on the minimized inference data and are happy to include these in the final paper if deemed valuable by the reviewers.
> > > > > >
> > > > > > Another point we'd like to stress is that, in our work, the entity, not the user, decides which questions to ask, thereby filtering the
> > > > > > information released.
> > > > > >
> > > > > > **2.** Your second question was related to our use of an English description for the problem objective formulation.
> > > > > >
> > > > > > While this choice was intentional to avoid additional notation, we have acknowledged your point and will include a mathematical description in response.
> > > > > >
> > > > > > **3.** Your third question related to hard label prediction.
> > > > > >
> > > > > > While hard label prediction is valuable in privacy-preserving ML --- since soft prediction can lead to leakage exploitable by adversaries (see, for instance, [1, 2, 3]) --- this restriction applies only to pure core feature sets (e.g., $\delta=0$). The presented approaches, however, are general and support "soft-label" predictions when $\delta > 0$.  Our experiments, utilizing varying uncertainty levels of $\delta$, further illustrate this.
> > > > > >
> > > > > > [1] https://arxiv.org/abs/2209.10732
> > > > > >
> > > > > > [2] https://arxiv.org/abs/2201.09370
> > > > > >
> > > > > > [3] https://arxiv.org/abs/2009.08559
> > > > > >
> > > > > > **4.** Your last point, was a suggestion to expand the scope to different contexts.
> > > > > >
> > > > > > We appreciate this suggestion!
> > > > > >
> > > > > > **5.** Finally, in response to your comment on related work, we emphasized that our work and setting significantly differ from the mentioned studies.
> > > > > >
> > > > > > We hope that this summary has clarified all of your doubts. Thank you again for your continued engagement.

---

> > > ### Author Response · Authors · 2023-08-14
> > >
> > > **Q3**: Hard labels are particularly useful in scenarios where discrete decisions are required by organizations such as banks or insurance companies. For instance, they may be used to determine whether a user qualifies for a loan.
> > >
> > > Consider the motivating example of a logistic regression model given by $1 \times \text{Job} - 0.5 \times \text{Loc} + 0.5 \times \text{Inc} \geq 0$. If some users have their Job value of 1, we can confidently predict that their hard label will always be 1. However, their soft label, or prediction score, depends on the unrevealed attributes Inc and Loc. For example, user A with Job = 1, Loc = -1, and Inc = 1 will have a score of $ \frac{1}{1 + \exp(-2)}$, while user B with Job = 1, Loc = 1, and Inc = -1 will have a score of $ \frac{1}{1 + \exp(0)}$.
> > >
> > > By estimating only hard labels, users can reveal fewer sensitive features without compromising the model's accuracy. This approach not only minimizes privacy but also reduces the burden on the bank or institution and saves time for the users that have to provide their data.
> > > This efficiency comes at the cost of providing less information, as no soft labels or prediction scores are given. This trade-off aligns with the well-known principle of the 'no free lunch' theorem.

---

### Official Review · Reviewer_dE9z · 2023-07-20

**Soundness:** 3 good
**Presentation:** 3 good
**Contribution:** 2 fair
**Rating:** 6
**Confidence:** 4

**Summary:**

This paper considers the problem of data minimization at inference time. Consider a set of features X which consists of public features Xp, and private features X\Xp. A model has been trained with all the features X, i.e., f(X). Now, the goal is to allow for inference revealing only a subset of the private features Xr and keep some of them unrevealed Xu.
In essence, X=Xp U Xr U Xu.


**Strengths:**

The paper introduces a new and interesting problem, i.e., data minimization during inference time.

They provide entropy-based measures to decide whether to include a feature or not.

They also provide theoretical guarantees and experimental results.

**Weaknesses:**

While the authors say this is the "first" work to do so, I think there are substantial similarities with the problem of feature selection or feature engineering as well as inference under missing data. The differences should be made clear with references.

The use of the term leakage can be a bit confusing since leakage is used here for the number of features revealed. But of course, the features can just be random noise and hence not leak anything, whereas one feature can also be much informative of everything and cause leakage. I think a different terminology should be used here since leakage sounds more like "information" rather than "number of features".

My most important concern: How does this approach compare with applying a local explanation framework and dropping the least important private features? For example, for each user, if you apply SHAP or LIME or even feature attributions, and then drop the private features with the least contribution for that local user. What performance would you get and how would that compare to your method?
I would be happy to increase my rating based on a discussion/comparison with just applying existing local explanations for this problem statement.

**Questions:**

See Weaknesses
COMMENT: After rebuttal, I increased my score by 2 points.

**Limitations:**

See Weaknesses (last point on my major concern)

---

> ### Author Rebuttal · Authors · 2023-08-08
>
> Thank you for your time and review. Below, we report our answers to your questions. Feel free to let us know if there are further questions or concerns and we'll be more than happy to elaborate.
>
> ### Q1
>
> Thank you for pointing out the apparent similarities between our work and existing problems in feature selection, feature engineering, and inference under missing data. We acknowledge that these are relevant domains, and a thorough understanding of how our work differs is crucial.
>
> *We have indeed addressed these differences in Section B (Related Work) of the Appendix*, where we detailed the distinctions between our approach and prior works. The space constraints in the main text prevented us from including this comparison there, but we ensured that the appendix provides a comprehensive examination.
>
> Should you require more specific insights or have any particular concerns about these comparisons, please let us know, and we'll be glad to address them.
>
>
> ### Q2
>
> Thank you for highlighting the potential confusion surrounding our use of the term "leakage" to describe the number of features revealed. Your observation about the possibility of one feature being highly informative, or features being mere random noise, is insightful.
>
> In our work, we specifically focused on the concept of leakage as the percentage of sensitive features a user needs to reveal. This notion aligns with privacy policies on data minimization, such as GDPR, and we found it to be an interpretable measure in our context. In all the datasets we explored, the corner case you mentioned did not occur.
>
> However, we do acknowledge your concern and agree that considering alternative terminology could help clarify our intentions. Additionally, the idea of reporting a noisy version of a sensitive feature as a privacy precaution is worth further exploration. Your feedback has prompted us to reflect on our terminology, and we appreciate the opportunity to clarify our approach. We will take into account your suggestion in our revision.
>
> ### Q3
>
> This is a great question. In our specific context of data minimization, applying SHAP or LIME to each individual would necessitate the revelation of *ALL* features at test time in order to quantify the contribution of each feature. This requirement contrasts with our goal of minimizing the number of sensitive features released at test time.
>
> Our method is aligned with preserving privacy by controlling the information revealed, and we've designed it to consistently improve over methods that selects features based on their importance (evaluated on the entire population). Our results across different datasets substantiate this improvement.
>
> We appreciate this thoughtful feedback, and it will certainly guide our further analysis and discussions in future revisions. We also hope you could reconsider your score, in light of our responses to your questions.

---

> > ### Author Response · Authors · 2023-08-13
> >
> > We wanted to reiterate our gratitude for your time and review and would like to check if you had any additional questions or comments. Many thanks!

---

> > > ### Comment · Reviewer_dE9z · 2023-08-14
> > > **Increasing score by 2 points**
> > >
> > > The rebuttal addresses my concern. For the final version, I would encourage the authors to (i) clarify the term leakage, and (ii) include an explicit discussion on applying local explanations.
> > > Increase: Borderline reject >> Weak accept

---

> > > > ### Author Response · Authors · 2023-08-15
> > > >
> > > > Thank you once again for your input. We look forward to enhancing our paper including your suggestions (i) and (ii).
> > > > Once again feel free to let us know if there are further questions and we'll be more than happy to elaborate.

---

### Official Review · Reviewer_ahUZ · 2023-07-31

**Soundness:** 3 good
**Presentation:** 2 fair
**Contribution:** 3 good
**Rating:** 6
**Confidence:** 3

**Summary:**

The authors propose that in a large number of application of machine learning reasonable model accuracy can be realized without the model having access to the entire feature set. This has implications for privacy and data-sharing, as a model that adaptively selects features to solicit would retain model performance even when minimal sensitive data is revealed to the model.

**Strengths:**

* The motivation for the paper is clear and important. The ability to reduce the amount of disclosed data is appealing and a thoughtful, timely contribution.

* The authors attempt to address obvious questions that may arise, specifically around the relationship between the input features being non-linear.


**Weaknesses:**

* the empirical evaluation of the model is somewhat weak. The authors do present results in terms of accuracy, however, there is no discussion as to what is the source of performance degradation, if any. It would be nice to see a more holistic evaluation that takes into account other aspects of the classifier performance, including ROC performance and calibration.

* For linear classifiers, computing the argmax over the estimated entropy is easy as it factorizes over the individual dimensions of the core set. However, this may not be the case when the relationship between features is non-linear. Can the authors comment on this. Specifically can the presented algorithm still be tractable if higher-order interactions between core-set of features considered?

* The current evaluation pipeline randomly assign attributes as un disclosed at random. I would recommend pre-specifying the disclosure rate based on certain notion of harm revelation of that feature would cause.

**Questions:**

The authors proposed a feature sampling strategy based on greedy selection on the Exected Information Gain. Such a strategy has shown benefit in other areas of machine learning especially in active learning. There could be potentially many other well motivated acquisition functions in the context of the problem the authors present. Have the authors considered other such possibilities? If yes, can the authors comment as to why Expected Information Gain (Entropy) was selected as the most reasonable choice.


The presented algorithm in its current stage does a one step-greedy lookahead which would work when the relationship between features is linear. Can the authors comment on tractability when looking at the power set of all potentia features that make the coreset?


I am willing to tweak my scores based on engagement with the authors and other reviewers in the discussion phase.

**Limitations:**

There are issues with model safety and trust when a model adaptively solicits information from a user. Specifically, can the authors comment on how such an approach might effect users trust in the system when the model seeks disparate information from disparate users?

I am not looking for a theoretical argument here, but more of a value judgement based discussion of potential implications of disparate treatment under such a model.

---

> ### Author Rebuttal · Authors · 2023-08-08
>
> Thank you for your time and review! If there are further questions or concerns and we'll be more than happy to elaborate.
>
> ### Empirical evaluation
> We appreciate the suggestion to include a more holistic view of the classifier's performance. Below, we address your concerns and provide additional insights.
>
> 1. **Trade-off between Data Leakage and Accuracy:** We indeed discuss the trade-off between data leakage (the number of features a user needs to reveal) and model accuracy in Section 3. Our approach is tailored to the user's willingness to reveal more or fewer features, leading to varying levels of accuracy. We believe this aspect of our work offers a nuanced understanding of how information revelation impacts predictions.
>
> 2. **Choice of Evaluation Metric:** We chose accuracy over the AUC metric for two main reasons:
> - AUC is generally designed for binary classification tasks. In our work, as elaborated in the Appendix, we also tackle multiclass classification problems.
> - Our primary concern is to obtain the model's hard prediction, which can be estimated with 100% confidence (delta = 0) even when not all sensitive features are revealed. This contrasts with soft-label prediction, where 100% confidence is impossible if some features remain unrevealed. Hard-label prediction requires users to reveal less sensitive information than soft-label prediction when considering minimum uncertainty or aiming for 100% confidence.
>
> 3. **Additional AUC Analysis:** We understand the importance of the AUC metric, and to address your concern, we have evaluated our F-score (with $\delta=0.2$) method against the Random and Feature Importance methods on Bank data. The results, shown in the table below, indicate similar AUC values to the baseline Importance, with the added benefit that users need to reveal fewer features (as detailed at the bottom of Figure 6).
>
> |   m  | F-score | Random | Importance |
> |------|---------|--------|------------|
> |  4   |  0.874  | 0.871  |   0.874    |
> |  5   |  0.874  | 0.872  |   0.875    |
> |  6   |  0.860  | 0.854  |   0.860    |
>
> ### Entropy computation
> We recognize that the complexity of computing the argmax over the estimated entropy becomes more challenging when we move from linear to non-linear models. Specifically, the ease of factorization over individual dimensions, as found in linear classifiers, no longer applies in the non-linear context. In our exploration, we observed that non-linear models do indeed incur more computational overhead but this increase in complexity is a trade-off we accept for the following reasons:
> - _Increased Accuracy_: Non-linear models are often capable of capturing more intricate relations within the data, leading to higher accuracy in general compared to linear ones.
> - _Higher-Order Interactions_: Despite the added complexity, our algorithm's ability to consider higher-order interactions between core-set features opens the door to more sophisticated and nuanced modeling. This approach may allow us to uncover relationships that linear models cannot.
>
> We believe that the potential gains in modeling capability justify this trade-off.
>
> ### Evaluation pipeline
> We agree that sensitivity varies by application and cultural context, and expert knowledge is often needed to make these determinations. For example, the disclosure of political orientation might have different implications in different regions.
>
> In our current evaluation, we opted for random assignment to maintain generality. However, we acknowledge that your suggestion could lead to a more nuanced evaluation. Your insight is appreciated, and we're open to further dialogue on this matter.
>
> ### Q1 (Expected information gain)
>
> Thank you for this insightful question.
> Indeed, we explored several alternative methods for our feature sampling strategy, including **(1)** an importance-based approach that prioritizes revealing more critical features first, and **(2)** a strategy based on the uncertainty of a feature given all revealed features. Our work compares the former in the main text and appendix.
>
> Note that approaches such as uncertainty score, do not account for the model's prediction, limiting their effectiveness, while the Expected Information Gain is able to capture the benefit of revealing a particular feature in the context of our problem.
> We believe that our choice, backed by its proven efficacy in areas like active learning, represents a well-motivated and robust solution for the data minimization context, but we remain open to exploring other methods.
>
>
> ### Q2 (Algorithm's details)
>
> As we elaborated in the main text (Lines 311 to 313), we did compare our proposed model with an _Optimal_ method that considers all possible subsets to choose the smallest coreset. Unfortunately, this exhaustive approach is not practical.
> Additionally, it poses a significant challenge in that it requires users to reveal all features at testing time, which is against the core motivation of this work.
>
> Our proposed strategy, in contrast, offers a more tractable solution, particularly when the relationship between features is linear, balancing efficiency with effectiveness.
>
> ### Q3 (Fairness)
>
> You raise a great point regarding trust and disparate treatment. While our current work primarily focuses on the trade-off between accuracy and data leakage, we recognize the importance of fairness and the potential implications of unequal information solicitation across different demographic groups.
>
> Indeed, this is subject of our ongoing investigation. Ensuring fairness by equalizing the amount of information revealed across groups is a promising direction, and it's one that we're actively exploring in our current work.
>
> We fully acknowledge that a thorough investigation of these ethical and social dimensions is needed.
>
> We hope these clarifications adequately address your concerns. We are committed to making any necessary adjustments and look forward to any further suggestions you may have.

---

> > ### Author Response · Authors · 2023-08-13
> >
> > We wanted to reiterate our gratitude for your time and review and would like to check if you had any additional questions or comments. Many thanks!

---

### Official Review · Reviewer_oWvc · 2023-08-01

**Soundness:** 3 good
**Presentation:** 3 good
**Contribution:** 2 fair
**Rating:** 6
**Confidence:** 4

**Summary:**

The work presents a method to increase privacy of ML predictions at test time by asking users to reveal fewer features. The proposed method asks for features that are maximally informative of the prediction outcome given the features seen thus far and stops when the prediction outcome can be decided. The method addresses the case of linear as well as non-linear thresholded classifiers assuming Gaussian-distributed features and a local Taylor approximation of predicted label probability. Experiments on multiple real datasets show reduction in data required for similar accuracy as baselines.

**Strengths:**

Writing is clear. The method is explained clearly step-by-step. I like the presentation of the results in the plots.

Method is simple and is shown to work on real datasets. The idea of selecting maximally informative features until decision is reached is natural. The idea is executed well.

Problem of reducing data required for a prediction is important to increase user privacy, so the method has practical significance.

**Weaknesses:**

On the writing, some algorithmic details can be improved like mentioning the entropy calculation and calculating the core features sets for \delta>0. Core feature sets can be defined more rigorously by mentioning the sources of randomness in the probability expression. Results from datasets other than Credit, which are presented in Appendix, can be summarised in the main text.

Reasons for the effectiveness of the method is not clear to me and are not sufficiently explored in experiments. This is needed given the success of simplifying assumptions on Gaussian distribution of features. Is this because of the dataset or model characteristics, or is due to the algorithm? Baselines such as removing all sensitive features might help check if dataset characteristics are the main contributor. An analysis of the examples which are predicted correctly without revealing features can shed more light.

Some related work is missing. See detailed remarks. The technical contributions in light of this work is unclear.

Theoretical analysis leaves many questions open e.g. the impact of approximations, what does optimal procedures look like, how much training data is needed to get significant privacy gains, how to extend the method to non-linear classifiers and high-dimensional data.

**Questions:**

Notation in Definition 1 is unclear. What does the probability of prediction f_\theta mean for unobserved variables X_U? Do we take expectation over different values of X_U? An example of the meaning of this probability would help me.
Similarly, how are the unrevealed variables in entropy calculations, e.g. in term A of equation (3), handled?

Suppose we remove all 5 of the sensitive features, does this have significantly lower accuracy? That is, does choosing the sensitive features matter at all for prediction.

How is entropy computed in equation 4? Please describe the Bayesian model used for the data.

Consider adding the related work on following topics and discuss whether these are applicable to the problem setting.
Active measurement of features e.g. Li and Oliva 2021 ‘Active Feature Acquisition with Generative Surrogate Models’ http://proceedings.mlr.press/v139/li21p.html
Dynamic measurement of features in time e.g. Chang et al. 2019 ‘Dynamic Measurement Scheduling for Event Forecasting using Deep RL’ https://proceedings.mlr.press/v97/chang19a.html
Feature pruning for causal effect estimation e.g. Makar et al. 2019 ‘A Distillation Approach to Data Efficient Individual Treatment Effect Estimation’ https://ojs.aaai.org/index.php/AAAI/article/view/4375

---
After the rebuttal

My remaining concern relates to the second question above. Reasons for the success of the method are not clear from the experiments and how they are presented. For instance, seeing the accuracy of a baseline which removes all sensitive features (minimum data leakage) will help contextualize all line plots (e.g. percentage improvement from such a baseline can be the y-axis). This helps answer how much of the success is due to the method's feature selection versus predictability of the public features in the datasets.
Further, a detailed analysis of the examples that are predicted correctly with < 2 sensitive features will be instructive (e.g. looking at whether these are the same set of features or personalized to the data point).
This said I appreciate baselines included by authors to check the selection criteria and experiments to check effect of linearity, which are also required for a newly proposed method.

**Limitations:**

Limitations on modeling assumptions and theoretical analysis are acknowledged.


## Minor comments, no response is aspected

Line 4 of the Algorithm 1 to check core features should be explained in detail for the case of \delta>0.

Please specify the assumption that input features are jointly Gaussian more prominently. e.g. in an Assumption environment, in the lines 177-180.

Please cite information processing inequality in Proposition 2 in main text.

Propositions 2 to 5 and Theorem 1 are known results so they can be mentioned in text or denoted as lemmas with references.

The statement in line 174 does not require pointing to Proposition 1. It holds because of the definition of entropy.

Please provide guidelines on how to use the Gaussian approximation for categorical features.

Introduction is nicely written. However, the goal / objective of the paper is repeated multiple times which can be removed to be concise.

---

> ### Author Rebuttal · Authors · 2023-08-08
>
> Thank you for your time and review!  Below, we report our answers to your questions. Feel free to let us know if there are further questions or concerns. and we'll be more than happy to elaborate. We also hope that you consider updating your score, if these replies answer your questions.
>
> **Q1: Notation**: In our context, since $X_U$ represents unobserved variables, we treat them as multivariate random variables following the conditional distribution $ P(X_U ∣ X_R=x_R) $. The model prediction $f_\theta$ is inherently a function of $X_U$ and, therefore, is also considered a random variable.
>
> To clarify with an example, if $P(X_U | X_R = x_R) = 0.8$, this implies that for all samples drawn from the distribution
> $x_U \sim P(X_U | X_R = x_R)$, 80% of the time you will observe the model prediction
> $f(X_R = x_R, X_U = x_U)$ resulting in a value of 1, and 20% of the time, the prediction will be 0.
>
> The unrevealed variables in entropy calculations, such as in term A of equation (3), are handled through similar probabilistic reasoning, where the conditional probabilities are leveraged to account for the uncertainty associated with these variables. We hope this clears up any confusion regarding the notation and handling of unobserved variables.
>
> **Q2: Clarification**: The impact of removing the 5 sensitive features depends on their importance relative to public features. If they're significant contributors, their removal could notably affect accuracy. In our specific experiments, all features contributed to prediction. Thus, designating some as sensitive and excluding them could lead to a drop in accuracy. The choice of sensitive features indeed matters in our case.
>
> **Q3: Entropy computation**: The entropy in Equation (4) quantifies the uncertainty in the model’s prediction for unobserved sensitive features $X_U$. It's computed by estimating the distribution $f(X_R = x_R, X_U)$ of the model responses with revealed variables $X_R$, which requires estimating the conditional distribution $P(X_U | X_R = x_R) from the training data. This can be done efficiently under a Gaussian assumption or less efficiently with a Bayesian neural network model.
>
> We detail the Bayesian model used to estimate these conditional densities from Line 198 to Line 205, adhering to standard Bayesian neural network training and inference.
>
> **Q4: Related Work**: We appreciate the reviewer for highlighting relevant works related to active measurement, dynamic measurement, and feature pruning. Please notice that we reported a detailed discussion of related work and connection with differential privacy, feature selection, and active learning, in Appendix B.
> Among the mentioned papers, we find the work by Li and Oliva (2021) to be the most closely aligned with our research. However, there are distinct differences.
>
> In our study, as demonstrated in the motivating example of Section 2 and the testing core feature set in Section 5.2, we show that users do not need to reveal all sensitive features to obtain a hard-label prediction with 100% confidence. Additionally, we provide an efficient algorithm that determines whether the current set of revealed features can ascertain the value of the model prediction with complete certainty.
>
> As for the other mentioned works, Chang et al. (2019) and Makar et al. (2019), we find their topics relevant but only tangentially connected to our problem setting. We assure you that we will include these related works and the nuanced discussions in our revised paper. Thank you for pointing us toward these resources.
>
> **Other comments**: Thank you for the detailed and insightful feedback. We sincerely appreciate your comments and will take them into account as we revise our paper.
>
> Regarding the use of categorical features, it is possible to employ a Bayesian network to estimate $P(X_U | X_R = x_R)$. However, it is essential to recognize that learning a Bayesian network can be a slow process, particularly when dealing with high-dimensional data. We would like to clarify that this challenge is not specific to our work but is a general concern in conditional density modeling of multivariate variables. The contribution of our work falls obviously beyond these constraints.
>
> We hope this addresses your concern, and we welcome any further questions or feedback.

---

> > ### Author Response · Authors · 2023-08-13
> > **Discussion**
> >
> > We wanted to reiterate our gratitude for your time and review and would like to check if you had any additional questions or comments. Many thanks!

---

> > ### Comment · Reviewer_oWvc · 2023-08-14
> > **After the rebuttal**
> >
> > I thank the authors for providing a detailed response to my questions. Most of my concerns except on evaluation are addressed. I increased my score to 6. Weak Accept. Overall I am more positive of the paper due to its contribution as defining the feature selection problem in a new context (privacy, test-time feature selection), and the simplicity and effectiveness of the method.
> >
> > My remaining concern relates to Q2 in the rebuttal. Reasons for the success of the method are not clear from the experiments and how they are presented. For instance, seeing the accuracy of a baseline which removes all sensitive features (minimum data leakage) will help contextualize all line plots (e.g. percentage improvement from such a baseline can be the y-axis). This helps answer how much of the success is due to the method's feature selection versus predictability of the public features in the datasets.
> > Further, a detailed analysis of the examples that are predicted correctly with < 2 sensitive features will be instructive (e.g. looking at whether these are the same set of features or personalized to the data point).
> > This said I appreciate baselines included by authors to check the selection criteria and experiments to check effect of linearity, which are also required for a newly proposed method.
> >
> > My concerns on entropy calculation, and the notation are addressed -> I would suggest explicitly naming the standard Bayesian techniques (e.g. from [10]) used in the implementation as it is an important detail. I would also suggest including the clarification on the random variable f_theta in the Notation paragraph since the notation is otherwise ambiguous. Related work can be more detailed by discussing the feature acquisition literature.

---

> > > ### Author Response · Authors · 2023-08-15
> > >
> > > Thank you for the positive assessment and for recognizing the novelty and significance of our work in terms of defining the concept of data minimization at inference time and its relation with privacy.
> > >
> > > Let us provide some additional details on our assessment:
> > >
> > > Firstly, we'd like to assure you that we have made a meticulous exploration of our results. This indeed included an analysis of instances that are accurately predicted with a minimal number of sensitive features. Even when restricted to k=1, we observed that the selected features are **not** uniform across the different users.
> > > We recognize the significance of this observation (thank you for your suggestion!) and will provide a more detailed explanation in the final version of our paper.
> > >
> > > Next, we agree with your recommendation to present our findings in the context of a minimum data leakage baseline. We have indeed conducted an evaluation under such conditions and it revealed a substantial decline in accuracy. We'll detail it in our revised manuscript, further substantiating our method's efficacy.
> > >
> > > Once again, thank you for your constructive feedback. We hope this addresses your last concern and that you could further champion our work.

---

### Decision · Program_Chairs · 2023-09-21

**Decision:**

Accept (poster)

**Comment:**

Reviewers engaged with this paper and identified strengths in its focus on data minimization and its technical contributions. Following the rebuttal period, the overall sentiment for the paper was "uniformly weakly positive": all five reviewers leaned towards acceptance, but there was no champion. Given the borderline scores and the potential for impact, I also reviewed the submission, reading the reviews, responses, and the original submission.

**My recommendation is to accept the paper.** This recommendation is based on a uniformly positive assessment from all 5 reviewers, the lack of fatal flaws in the work, the broader need for technical work on data minimization in the literature, and the potential impact of this work provided that the authors can carry out some simple revisions. Below, I list the high-level strengths and weaknesses I weighed and include my suggested changes for the camera ready.

### Evaluation

**Strengths**

- Studies an emerging topic in the privacy (e.g., data minimization). This is a valuable task that deserves more attention in the literature.

- Proposed methods to train classifiers that minimize features at test time has several valuable implications (e.g., to minimize sensitive features in healthcare applications, or to minimize features that are costly to acquire)

**Weaknesses**

- Proposed methods "minimize features requested at prediction time" using "entropy" as a proxy. This issue is glossed over in the work and merits further attention.

- Motivation: The paper motivates the need for data minimization in a way that is overly abstract. This would be stronger with a few specific examples.

- Presentation: The paper suffers from a set of baffling decisions that make it difficult to read and evaluate. Methods are developed in standalone sections that also include experiments. Related work is deferred to an Appendix. Captions for the figures are short and uninformative. I have included suggestions below to address these issues.

### Proposed Revisions

### 1. Explicit Discussion of the Use of Entropy as a Surrogate.

I agree with ZsPj in that the greatest weakness of this work is the fact that the proposed methods use entropy as a surrogate measure. This decision could backfire in a harmful way ([see e.g., comments by ZsPj](https://openreview.net/forum?id=cZS5X3PLOR&noteId=0vu5nbVQv7)). It can also ability to apply the approach to analogous methods that minimize data with respect to other metrics (see e.g., the comments of [ddIE](https://openreview.net/forum?id=cZS5X3PLOR&noteId=5jHf3H9W6Q)).

IN this case, I did not find the authors' response to ZsPj convincing or commensurate with the concerns surrounding this issue. However, I decided to recommend acceptance as the experiments suggest that the proposed methods can work effectively. Given the positive results, I recommend a number of changes to improve clarity and establish soundness:

1. Explicitly state the potential limitations of the design choice in the main text (e.g., by using a Remark). Note that the assumption was only caught by ZsPj but missed by 4 of 5 reviewers.

2. Explicitly characterize when you would expect to run into failure modes as a result of the surrogate. This can be done, for example, by referring to related work, developing counterexamples, and evaluating in the experiments.

3. State the potential limitations in the conclusion. Discuss how to deal with it in practice, and how it might affect the design of future work.

### 2. Strengthen Motivation

If accepted, the paper will be well-positioned to inspire future work on this topic. To this end, I would advise the authors to articulate a clear and convincing case for data minimization -- with tangible examples and references to policies and laws.

As stands, the motivation for data minimization in the introduction is weak because it is too abstract. Consider the following statement from the introduction:

> This practice not only presents significant privacy risks for users but also burdens companies and organizations with an extensive human effort to verify the accuracy of disclosed information (e.g., auditing in finance operations).

Who is the "user" and "the organization?" What is the "burden"? What are the "privacy risks?" Has this actually happened or is this some narrative? The text would be far more convincing if it could update sentences like these to paint a clear picture to, e.g., an undergraduate student. Point to actual instances and regulations, naming users, organizations, and explaining the "burden" in precise terms.

### 3. Restructure for Readability

The paper would be far easier to read, skim, and understand if you could make a few changes.

- Integrate Related Work Section: I strongly recommend that the authors include a Related Work section in the paper. As mentioned by ddiE /  ZsPj, the paper overlooks a number of recent work related to data minimization. I would recommend the work reference these papers clearly and plainly to highlight a broader stream of research on this topic, and to cite principles and laws beyond legal requirements beyond those in the GDPR. The relationship between the current work and related papers should be *crystal clear*.

- Separate Methods from Evaluation: The last 3 sections of the paper should be restructured. I would recommend bundling them all into a single section with subsections ("Scoring Features", "Linear Alg", "Non-Linear Alg"). The experiments should be placed in a standalone section. The current structure doesn't allow for easy comparison between the algorithms and wastes space (leading to e.g., short captions and related work in the Appendix)

- Figure 1: This is a missed opportunity to clearly convey your problem and proposed solution. It should be developed into a full-width Figure so that readers who skim the paper can quickly understand your setting, and readers who read the paper can understand the order of operations. Note that [dE9z](https://openreview.net/forum?id=cZS5X3PLOR&noteId=IEV7a0sea0) would not have asked questions about the LIME/SHAP attribution method had there been a clear description of what you are trying to do.

- Captions: The captions for all figures in this work are too short and uninformative (e.g., "Figure 6 Neural Network Classifiers"). These captions should be updated to explain what you are showing and why.